# Robust Weight Imprinting: Insights from Neural Collapse and Proxy-Based Aggregation

**Justus Westerhoff**                                                                 *justus.westerhoff@bht-berlin.de*
*DATEXIS, Berliner Hochschule für Technik (BHT), Germany*

**Golzar Atefi**
*DATEXIS, Berliner Hochschule für Technik (BHT), Germany*

**Mario Koddenbrock**
*KI Werkstatt, Hochschule für Technik und Wirtschaft Berlin (HTW), Germany*

**Alexei Figueroa**
*DATEXIS, Berliner Hochschule für Technik (BHT), Germany*

**Alexander Löser**
*DATEXIS, Berliner Hochschule für Technik (BHT), Germany*

**Erik Rodner**
*KI Werkstatt, Hochschule für Technik und Wirtschaft Berlin (HTW) & Merantix Momentum, Germany*

**Felix A. Gers**
*DATEXIS, Berliner Hochschule für Technik (BHT), Germany*

**Reviewed on OpenReview:** *https://openreview.net/forum?id=duU11BnQ3Y*

## Abstract

The capacity of foundation models allows for their application to new, unseen tasks. The adaptation to such tasks is called transfer learning. An efficient transfer learning method that circumvents parameter optimization is imprinting. The conceptual differences between studies on imprinting form the basis of our systematic investigation. In this work, we propose the general `IMPRINT` framework, identifying three main components: generation, normalization, and aggregation. Through the lens of this framework, we conduct an in-depth analysis and a comparison of the existing methods. Our findings reveal the benefits of representing novel data with multiple proxies in the generation step and show the importance of proper normalization. Beyond an extensive analytical grounding, our framework enables us to propose a novel variant of imprinting which outperforms previous work on transfer learning tasks by 4%. This variant determines proxies through clustering motivated by the neural collapse phenomenon – a connection that we draw for the first time. We publicly release our code at `https://github.com/DATEXIS/IMPRINT`.

## 1 Introduction

In machine learning applications, training models from scratch is often not viable due to limitations in data and compute. A popular solution is to apply transfer learning (Bengio, 2012; Yosinski et al., 2014) based on foundation models (`FMs`) (Bommasani et al., 2021) that are pre-trained on a large amount of data. To tune a `FM` to a novel task, e.g., for classification, a common procedure is to freeze the model parameters and replace the output layer with a new head. A particularly simple method for implementing such a new head was proposed by Qi et al. (2018) and coined *imprinting*.

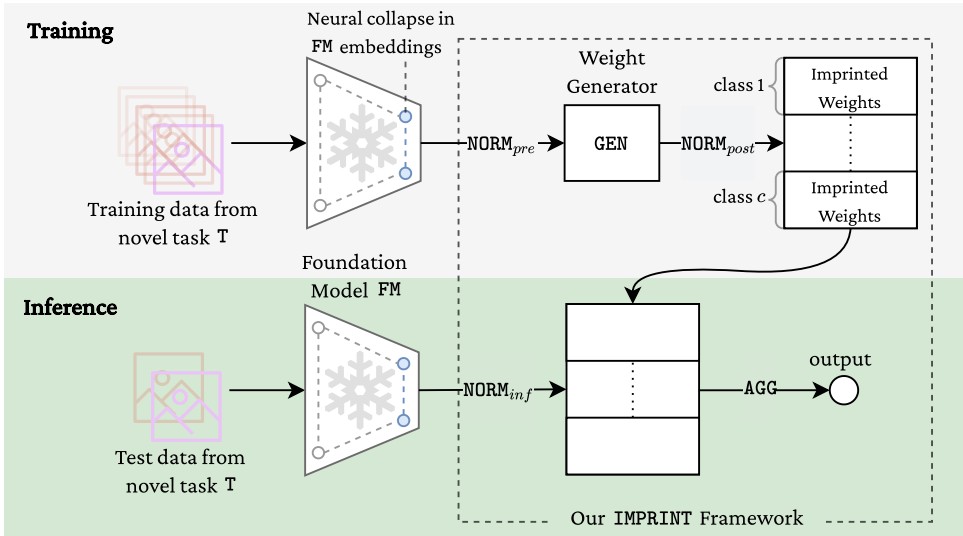

**Figure 1:** Overview of our `IMPRINT` framework. The foundation model `FM` is frozen and shows neural collapse. The weight generator (`GEN`) uses training data from a novel task `T` to consecutively generate one or more weight vectors (proxies) per class $1, \ldots, C$ in `T`. In inference, the final output for the test data in `T` is computed by an aggregation (`AGG`) mechanism. Embeddings and generated weights are normalized according to $\texttt{NORM}_{pre}$ and $\texttt{NORM}_{post}$, respectively. During inference, embeddings are normalized according to $\texttt{NORM}_{inf}$.

**Imprinting.** In the original work by Qi et al. (2018), the last-layer weight vector of a novel class is set to the normalized average of its scaled embedding vectors, i.e., its class mean. These class means are representatives of the classes, which we denote as *proxies*. In general, we refer to imprinting as efficient learning methods without the need for cross-class statistics or gradient-based optimization. A plethora of studies have emerged surveying this technique by adding complexity and adaptability (Gidaris & Komodakis, 2018; Siam et al., 2019; Passalis et al., 2020; Khan et al., 2021; Li et al., 2021; Cristovao et al., 2022; Yan et al., 2023). Despite its many adaptations, imprinting lacks a systematic comparison that unifies them. Understanding its variations could unlock greater efficiency across many fields, making the method even more versatile and impactful.

**Applications.** In particular, while in some practical applications, accuracy is prioritized over computational efficiency, the latter does become a critical requirement in scenarios where computational resources are severely constrained, e.g., in the chemical and polymer processing industries. Here, battery-powered edge compute is essential and frequent retraining or large-scale optimization is infeasible. Imprinting has proven particularly effective in these contexts of edge-embedded devices (Passalis et al., 2020). For instance, Zhu et al. (2022) implement a vision-based robotic force grasping with a variable-stiffness gripper that can safely handle both fragile and heavy objects by rapidly adapting to novel categories without retraining (Continual Learning (CL) setting). Industrial adoption also underscores this trend with Google's Coral Edge TPU including an `ImprintingEngine` API (Coral), that allows users to add new classes from a few examples without recompiling the model. Very recently, (Belal et al., 2025) apply imprinting to spectrogram embeddings from IMU/EMG gait data, achieving efficient classification in low-data Human Activity Recognition (HAR) tasks. Apart from that, Janson et al. (2022) extend imprinting to CL, establishing a simple baseline with competitive performance compared to more sophisticated state-of-the-art CL algorithms.

**Framework.** We present `IMPRINT`, a framework that enables a comprehensive analysis of existing imprinting techniques. More precisely, we generalize prior work by decomposing imprinting into three principal steps (see fig. 1). During generation (`GEN`) of weights, the method selects representative data samples and generates one or more weight vectors (proxies) per class. Normalization (`NORM`) is crucial, as the generated weights need to be balanced. Aggregation (`AGG`) entails the computation of the final output, e.g., a class label. The computational efficiency of imprinting allows us to perform a large number of experiments. Through `IMPRINT`, we are able to propose a novel, best-performing imprinting strategy using multi-proxy weight imprinting in combination with $L^2$ normalization, outperforming previously studied methods, as depicted in fig. 2.

**Neural Collapse.** When neural networks are trained to reach near-zero loss, their penultimate-layer embeddings collapse to the class means (Papyan et al., 2020; Zhu et al., 2021). We investigate this phenomenon as a potential explanation for when and why imprinting works. Our analysis proves that there exists a relationship between a measure of neural collapse and the success of imprinting. Since quantification of this phenomenon is possible in a post-hoc fashion over the `FM` features, we believe that these insights will contribute to further development of imprinting methods, as well as, training regimes of `FMs` that are more suitable for transfer learning via imprinting.

**Contributions.** In summary, our main results and contributions are:

- We deconstruct weight imprinting into the `IMPRINT` framework composed of generation, normalization, and aggregation, and discuss variations for each of them, while identifying prior work as special cases (section 3.1). To the best of our knowledge, we are the first to conduct a comprehensive analysis of imprinting to this scale (section 4).
- We present a new, outperforming imprinting method utilizing $k$-means clustering for weight generation (section 5.1) and show its benefits in certain low-data regimes (section 5.2).
- As far as we are aware, we are the first to identify a relationship between the degree of neural collapse and imprinting success (section 5.3).

## 2 Related Work

**Imprinting and Low-Data Regimes.** Weight imprinting is implemented by setting the final layer weights for the novel classes to the scaled average of the embedding vectors of their training samples and was first introduced by Qi et al. (2018) for the few-shot learning scenario. The authors find that for up to 20 samples, using a combination of imprinting and fine-tuning outperforms other state-of-the-art methods, including nearest neighbor algorithms. In contract, in our work we do not limit the number of samples and perform no fine-tuning on the imprinted weights to maintain efficiency.

Imprinting has been applied to object detection (Li et al., 2021; Yan et al., 2023), multi-label classification (Khan et al., 2021), semantic segmentation (Siam et al., 2019), and combined with an attention mechanism to generate weights for the novel classes in a few-shot classification task (Gidaris & Komodakis, 2018).

Hosoda et al. (2024) apply imprinting using quantile normalization to ensure statistical similarity between new and existing weights. We consider this as one normalization scheme in our framework. Zhang et al. (2021) apply imprinting in chest radiography for detection of COVID-19 and find that it yields better results than joint gradient descent training of all classes when only few samples are available. They speculate whether normalization is a constraint in their imprinting model.

Before the era of deep learning, Mensink et al. (2013) analyze the transferability of hand-crafted image features. They use a "nearest class multiple centroids" (NCMC) classifier with multiple proxies generated from a $k$-means clustering algorithm. In combination with metric learning, they compare favorably against the $m$-nearest neighbor algorithm. Our work, on the other hand, highlights efficient transfer learning provided by foundation models.

**Transfer Learning.** Using embedding vectors from pre-trained models is a simple and widely used transfer learning approach, established in the seminal works on computer vision (Donahue et al., 2014) and natural language processing (Devlin et al., 2019). Kornblith et al. (2019) show that pre-training performance of a model is highly correlated with the performance of the resulting embedding vectors in downstream tasks. In addition, Huh et al. (2016) provide insights into the required quality of pre-training data. Our work is orthogonal to these studies, since we focus on studying weight generation, normalization, and aggregation techniques applied later on for new task adaptation.

**Continual Learning (CL).** Although we investigate transfer learning scenarios, we review the imprinting applications and results from CL. Rebuffi et al. (2017) dynamically select a subset of examples for each class and update internal representations via gradient descent. They use a nearest mean classifier (NMC)

**Table 1 & Figure 2:** Previously studied imprinting strategies are special cases within `IMPRINT`. We evaluate 12 different classification tasks `Ts` derived from *MNIST*, *FashionMNIST*, and *CIFAR-10*, each with 10 classes or subsets thereof, and 4 pre-trained models `FMs` (`resnet18`, `resnet50`, `vit_b_16`, `swin_b`). The proposed configuration ("Ours") derived from `IMPRINT` outperforms previous work across `FMs` and `Ts` by a large margin with statistical significance, as confirmed by the critical difference (CD) diagram below. Since absolute accuracies vary substantially across tasks and models (as reflected by the large standard deviations (std)), this rank-based aggregation is used for fair comparison. Here, $k = 20$ is used, highlighting the gain of using multiple proxies per class. For reference, the gray row reports an oracle method that uses cross-class feature statistics to generate weights (see appendix A.5). It is not an imprinting method and therefore not directly comparable to the imprinting-based approaches above. Nonetheless, the results indicate that our method substantially narrows the gap between single-proxy `mean` imprinting and this oracle baseline.

| Work | $\text{NORM}_{pre}$ | GEN | $\text{NORM}_{post}$ | $\text{NORM}_{inf}$ | AGG | Avg. acc. $\%\ \pm$ std |
|------|------|------|------|------|------|------|
| Qi et al. (2018) | L2 | mean | L2 | L2 | max | 86.79 $\pm$7.83 |
| Hosoda et al. (2024) | none | mean | quantile | none | max | 82.90 $\pm$12.87 |
| Janson et al. (2022) | none | mean | none | none | 1-nn | 86.64 $\pm$7.88 |
| Ours | L2 | k-means | L2 | L2 | max | **91.06** $\pm$6.21 |
| Oracle | none | least-squares | none | none | max | 94.54 $\pm$5.00 |

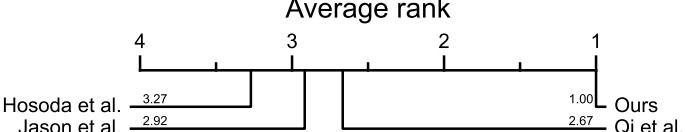

with respect to the saved examples. Janson et al. (2022) use an NMC classifier as well and achieve good performance on CL benchmarks without any fine-tuning of the embeddings. However, they do not investigate the effect of normalization and using multiple proxies.

Findings of Prabhu et al. (2023) show that a simple, approximate $m$-nearest neighbor classifier outperforms existing methods in an Online CL setting when all data can be stored. In our work, however, we compare imprinting all data to a limited number of more representative proxies striving for efficiency.

**Neural Collapse (NC).** The phenomenon of NC was identified by Papyan et al. (2020) and refers to the convergence of the last-layer weight vectors to class means. It was shown that, regardless of the loss function, optimizer, batch-normalization, or regularization, NC will eventually occur (provided the training data has a balanced distribution) (Zhu et al., 2021; Han et al., 2022; Kothapalli, 2023). Nevertheless, complete neural collapse is practically unrealistic (Tirer et al., 2023). In transfer learning, Galanti et al. (2022) show that NC occurs on new samples and classes from the same distribution as the pre-training dataset, highlighting the usability of foundational models in such scenarios. In our work, we expand the survey on NC by experimenting with out-of-distribution classes belonging to different datasets and linking their degree of collapse to the success of certain imprinting strategies.

## 3 Methods

### 3.1 `IMPRINT`

In order to find out how to best and efficiently set the classifier weights of a foundational model `FM` in new, previously unseen tasks `T`, we create the `IMPRINT` framework (see fig. 1) that generalizes previous work, specifically all the methods that work without access to cross-class statistics and gradient-based training. Thereby, we unify all the existing imprinting strategies described in section 2.

**Overview.** We analyze the effect of weight generation (`GEN`), normalizations (`NORM` = {`NORM`$_{pre}$, `NORM`$_{post}$, `NORM`$_{inf}$}), and aggregation (`AGG`). The `IMPRINT` framework depicted in fig. 1 consists of three main building blocks: a foundation model `FM`, a weight generator `GEN`, and extendable classifier weights that are imprinted. The `FM` remains frozen throughout the experiments. It receives data from `T` as inputs and produces embedding vectors. The training process generates weight vectors for each of the $C$ classes in `T`. Hereby, embeddings from the `FM` are normalized before the generation (`GEN`) step according to `NORM`$_{pre}$. The generated weight vectors per class are referred to as *proxies*, prototypes, or representatives (Movshovitz-Attias et al., 2017; Snell et al., 2017; Yang et al., 2018; Yu et al., 2020). These proxies are normalized according to `NORM`$_{post}$. As in the work by Qi et al. (2018), we do not use bias values. To classify the test data in `T` during inference, it is first embedded by the `FM`, normalized according to `NORM`$_{inf}$, and finally aggregated by `AGG`, resulting in a predicted class label.

**Generalization of Previous Methods.** Previously proposed imprinting methods can be defined as special cases of our framework. We design `IMPRINT` such that every method can be defined by a single combination of `GEN`, `NORM`, and `AGG`, of which we inspect all possible combinations.

**Weight Generation (`GEN`).** The purpose of `GEN` is to determine how the embeddings of the training data in `T` are used to form the new weights. In contrast to Qi et al. (2018) which only incorporates one proxy per class (the mean), we add flexibility by allowing each class to have multiple proxies as in Mensink et al. (2013) and enable non-linear classification. We denote the number of proxies per class as $k$, ranging between 1 and the number of samples, and investigate the following operations **conducted per class** to generate those:

- `all`: All embeddings (denoted as $k =$ all).
- `k-random`: $k$ random embeddings.
- `mean`: The mean of all embeddings.
- `k-means`: $k$-means cluster centers using `KMeans` from *sklearn* (Pedregosa et al., 2011). $k = 1$ is the same as `mean`.
- `k-medoids`: $k$-medoids cluster centers using `KMedoids` from *sklearn*.
- `k-cov-max` (covariance-maximization): Top $k$ embeddings by covariance (in code: `proxies = embeddings[torch.argsort(torch.sum(torch.cov(embeddings), dim=0), descending=True)[:k]]`).
- `k-fps` (farthest-point sampling): Iteratively selecting $k$ embeddings, such that it maximizes the distance from already selected ones (starting with a random sample).

We choose this diverse list of methods to cover a wide range of approaches, ranging from heuristics (e.g., `k-fps`) to more complex algorithms (e.g., `k-means`). Note that only `mean` and `k-means` generate proxies beyond existing samples by producing synthetic cluster centers in embedding space. In comparison, `k-medoids` must choose actual samples as cluster centers (analogous to a median). None of these methods use cross-class statistics or gradient-based optimization. We also note an analogy to associative memory models, interpreting imprinting as a memory update process following a covariance rule, as detailed in appendix A.3.

**Normalization (`NORM`).** The main reason for applying normalization is to allow each embedding and weight vector to contribute equally on the same scale. The modes we allow are no normalization (`none`), $L^2$ normalization (`L2`), and quantile normalization (`quantile`).

`L2` normalization can be applied to embeddings before `GEN` via `NORM`$_{pre}$, to the generated weights via `NORM`$_{post}$, and to embeddings in inference via `NORM`$_{inf}$. In any case, the vector is $L^2$-normalized by dividing it by its Euclidean length $\| \cdot \|_2$.

`quantile` normalization (Amaratunga & Cabrera, 2001; Bolstad et al., 2003) can only be applied to generated weights. This non-linear operation distributes weights equally. Recall that if more than one class is contained in `T` ($c > 1$), `GEN` is performed for each class consecutively, and the reference distribution changes accordingly. In particular, for the first class there is no reference distribution to map to. This is different from Hosoda et al. (2024), where new weights are matched to the distribution of the original classifier weights of the `FM`.

Since we do not consider the classes used for pre-training the `FM` and especially do not assume access to their last-layer weights, this is not possible in our scenario.

**Aggregation (`AGG`).** There are various ways to use the generated weights (proxies) per class during inference, especially when $k > 1$. We focus on two different modes, `max` and $m$-nearest neighbor (`m-nn`). The former, `max`, computes the inner product of the input embedding and the imprinted weights and outputs the class label with the maximum activation. The latter, `m-nn`, uses the class weights as keys and the embeddings as values, and chooses the final winning output class via the $m$-nearest neighbor algorithm. The `m-nn` voting is weighted by the inverse of the Euclidean distances to their nearest neighbor, turning it into weighted majority voting.

We also experimented with alternative distance functions (Cosine, Manhattan) and uniform weighting, but found differences well within statistical noise. Using the Mahalanobis distance proved computationally prohibitive. More elaborate voting mechanisms, such as learned top-$k$ attention or entropy-based filtering, are conceivable, but extend well beyond the minimalist imprinting paradigm and therefore left for future work.

Note that `max` is the same as `1-nn` in the case of `L2` for $\text{NORM}_{post}$, since for any fixed embedding vector $\mathbf{v}$ and variable proxy $\mathbf{w}$, the argmin of $\|\mathbf{v} - \mathbf{w}\|^2 = \|\mathbf{v}\|^2 - 2\langle\mathbf{v}, \mathbf{w}\rangle + \|\mathbf{w}\|^2$, calculated by `1-nn`, is the same as the argmax of the inner product $\langle\mathbf{v}, \mathbf{w}\rangle$ calculated in `max`.

## 3.2 Quantifying Neural Collapse

Neural collapse (NC) (Papyan et al., 2020) characterizes the state of the features produced by a classification neural network after training to near zero training loss. Namely, the learned embeddings of each class converge, i.e., *collapse*, to their class means. These globally centered class means and classifier weights form a simplex equiangular tight frame (ETF) – a collection of equal length and maximally equiangular vectors, that maximize the between-class variability. This results in an optimal linearly separable state for classification. In fig. 3 (left), we illustrate the collapse of a `FM` on its pre-training data. The newly arrived data `T` from a different dataset is distributed more unevenly across the embedding space (right).

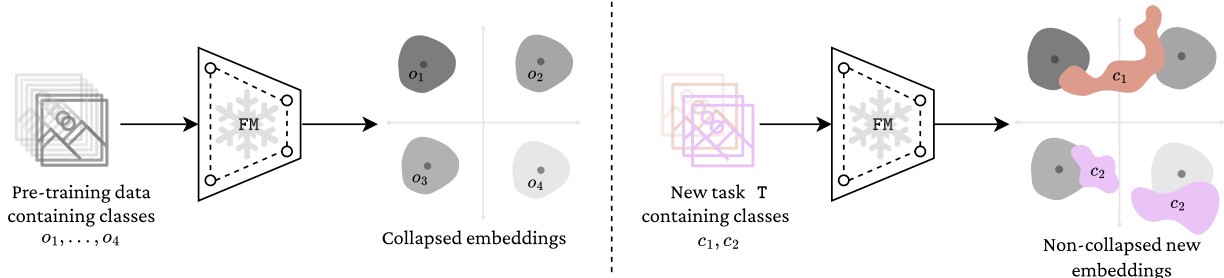

**Figure 3:** Left: The embeddings of the pre-training data, after being used to train the foundation model `FM`, show neural collapse, as each class $(o_1, \ldots, o_4)$ is evenly separated in space and accumulates around their respective class means. Right: For a novel task with classes $c_1, c_2$ (pink and brown) scatter around the collapsed pre-trained classes (gray).

Two important characteristics of NC are **variability collapse**, i.e., the within-class variability of the penultimate-layer embeddings collapses to zero, and **convergence to nearest-mean-classification**. We focus on variability collapse ($\mathcal{NC}_1$) as in Zhu et al. (2021). Given a foundation model `FM` and a finite dataset containing $C$ classes with (for simplicity) $N$ samples per class, we use its $l$-dimensional embeddings $\{\mathbf{h}_{c,i}\}_{1 \le c \le C,\, 1 \le i \le N}$ given by `FM` to define the global feature mean $\mathbf{h}_G := \frac{1}{CN}\sum_{c=1}^{C}\sum_{i=1}^{N}\mathbf{h}_{c,i}$, class means $\overline{\mathbf{h}}_c := \frac{1}{N}\sum_{i=1}^{N}\mathbf{h}_{c,i}$, the within-class covariance matrix $\boldsymbol{\Sigma}_W := \frac{1}{CN}\sum_{c=1}^{C}\sum_{i=1}^{N}(\mathbf{h}_{c,i} - \overline{\mathbf{h}}_c)(\mathbf{h}_{c,i} - \overline{\mathbf{h}}_c)^\top$, and the between-class covariance matrix $\boldsymbol{\Sigma}_B := \frac{1}{C}\sum_{c=1}^{C}(\overline{\mathbf{h}}_c - \mathbf{h}_G)(\overline{\mathbf{h}}_c - \mathbf{h}_G)^\top$ to finally compute

$$\mathcal{NC}_1 = \tfrac{1}{C}\text{trace}(\boldsymbol{\Sigma}_\mathbf{W}\boldsymbol{\Sigma}_B^+), \tag{1}$$

where $^+$ symbolizes the pseudo-inverse.

Based on eq. (1), an $\mathcal{NC}_1$ score closer to zero signifies a higher collapse. In contrast, an increase in multi-modality of data leads to a higher $\mathcal{NC}_1$ score (as analyzed in fig. 11a). Note that this measurement is not independent of the embedding dimension $l$ and the number of classes $C$. According to NC, imprinting the mean, as originally done by Qi et al. (2018), is best when $\mathcal{NC}_1$ is small. We claim that when the data is not fully collapsed (as is often the case in practice), the scale of $\mathcal{NC}_1$ could guide the proxy generation method, e.g., having multiple proxies $k > 1$ per class. We investigate this in section 5.3.

### 3.3 Significance Testing with Critical Difference Diagrams

Our experiments compare large numbers of imprinting configurations across tasks with substantially different achievable accuracy levels. To obtain scale-independent statistical comparisons, we base our analysis on ranking (dis-)agreements rather than raw accuracy. Non-parametric rank-based tests are well suited for this setting. Following Demšar (2006), we use the Friedman test and Wilcoxon signed-rank tests because they do not assume normality or homoscedasticity, and can be applied uniformly to any evaluation metric. Empirical evidence in Demšar (2006) further suggests that these tests have good power in classifier comparison scenarios.

Concretely, for each evaluation instance (i.e., foundation model `FM` and task `T`) we compute the classification accuracy of all configurations and assign ranks (rank 1 = highest accuracy). This yields a matrix of ranks with one row per configuration and one column per evaluation instance. On this matrix we first apply the Friedman test to assess whether there are any overall differences between configurations. If the Friedman test rejects the global null hypothesis at significance level $\alpha = 0.05$, we proceed with a post-hoc, pairwise analysis. Specifically, for every pair of configurations we run a two-sided Wilcoxon signed-rank test across evaluation instances. The resulting $p$-values are corrected for multiple comparisons using the Holm step-down procedure (Holm, 1979).

Critical difference (CD) diagrams summarize both the average performance and the significance structure. The horizontal axis shows the average rank of each configuration across all evaluation instances, with lower ranks indicating better performance. Two configurations are connected by a thick horizontal line if their Wilcoxon–Holm comparison does not show a significant difference.

## 4 Experimental Setup

**Foundation Models `FM`.** We use `resnet18`, `resnet50` (He et al., 2016), `vit_b_16` (Dosovitskiy et al., 2021), and `swin_b` (Liu et al., 2021) as `FM`s, two CNN-based and two Transformer-based architectures. All four models are pre-trained on *ImageNet*-1K (ILSVRC 2012) (Deng et al., 2009). To generate the embeddings, we use PyTorch's *torchvision* models.

**Tasks `T`.** We analyze multi-class classification scenarios without separating base and new classes, instead focusing on all classes within a novel `T` at the same time. To investigate the effect of the number of samples given, we look at `n-shot` ($n \in \mathbb{N}$) scenarios. For that, we randomly pre-sample the training data of `T` to $n$ samples per class – transitioning into the low-data regime.

To find out the best imprinting strategy within our `IMPRINT` framework, we focus on tasks `T` created from the datasets *MNIST* (Deng, 2012), *FashionMNIST* (Xiao et al., 2017), and *CIFAR-10* (Krizhevsky et al., 2009), each containing 10 classes. We mainly focus on the three `T` containing all ten classes. Furthermore, we look at smaller tasks only containing classes $\{0, 1, 2\}$, and the two tasks containing classes $\{1, 3, 5, 7, 9\}$ resp. $\{0, 2, 4, 6, 8\}$. This random selection of $3 \cdot 4 = 12$ tasks adds variation to our evaluations.

**Neural Collapse.** In the analysis of neural collapse (NC), we also look at the `FM`s' pre-training data (*ImageNet*). As its test set is not available, we use its validation set in $\mathcal{NC}_1$ computations. Furthermore, for *ImageNet*, we relabel data by combining multiple classes into one label to simulate multi-modal class distributions for an in-depth NC analysis. These tasks are called "$d$ in 1", $d = 1, \ldots, 10$, each containing 10 different labels. More precisely, we take 100 random classes from *ImageNet* and sequentially map the first $d$ to label 1, the second $d$ to label 2, etc., until we reach 10 distinct labels. See fig. 4 for a simplified illustration.

Moreover, we construct a new out-of-distribution, non-collapsed dataset by merging all classes from four digit datasets: *MNIST*, *MNIST-M* (Ganin et al., 2016), *SVHN* (Netzer et al., 2011), and *USPS* (Hull, 1994). The resulting dataset, which we refer to as *CombiDigits*, exhibits reduced collapse due to the greater distributional diversity within each class. To ensure scale invariance in covariance-based NC measurements, all embeddings are $L^2$-normalized before computing $\mathcal{NC}_1$.

**Scale.** In total, we ran approximately $500\,000$ experiments, varying imprinting components, foundation models, tasks, and seeds. This is feasible with minimal effort as imprinting is a highly efficient method: core steps such as weight generation (`GEN`), normalization (`NORM`), and aggregation (`AGG`) are linear in dataset size or number of proxies and parallelize efficiently within each step. Details on experimental infrastructure and parallelization are provided in appendix A.6.

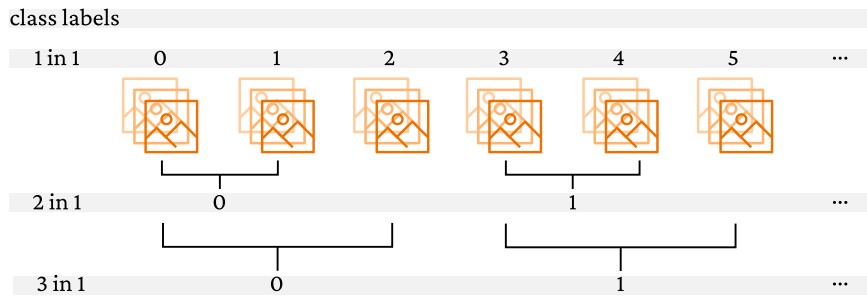

**Figure 4:** Combining multiple classes into one to create tasks with multi-modal class distributions. Simplified example for "$d$ in 1", $d = 1, 2, 3$, with only six (instead of 100) original classes.

**Evaluation.** Unless stated otherwise, we report the median test accuracy over three different seeds. In sections 5.1 and 5.2, we evaluate the imprinting performance by varying the `FM` (4) and `T` (12) and report average accuracy and standard deviation (std) across these. Due to the heterogeneity of models and tasks, large std are expected. Therefore, overall method comparisons are based on ranks rather than absolute accuracies. Methods are ranked by their median accuracy, yielding $4 \cdot 12 = 48$ potentially different ranks. We report the average rank and assess statistical significance of ranking (dis-)agreements using critical difference (CD) diagrams as explained in section 3.3. The code used to generate these diagrams is inspired by Ismail Fawaz et al. (2019).

In experiments with neural collapse (section 5.3), we investigate the four `FM`s on 100 random *ImageNet* tasks with remapped labels (as explained above) and the four tasks containing all of *MNIST*, *FashionMNIST*, *CIFAR-10*, and *CombiDigits*, respectively.

## 5 Results

Our main experimental insights are: **1.** Our `IMPRINT` framework generalizes previous methods, and we find a new superior imprinting strategy (section 5.1). **2.** We show that our novel strategy is even beneficial in low-data regimes with as little as 50 samples per class (section 5.2). **3.** We identify a correlation between imprinting success utilizing multiple proxies and a measure of neural collapse (section 5.3).

### 5.1 Best Imprinting Strategy

We provide a comparison between existing memory-constrained methods used for imprinting on foundation models in fig. 2, namely, the work by Qi et al. (2018); Hosoda et al. (2024); Janson et al. (2022), as well as a novel, best-performing configuration ("Ours") that results from `IMPRINT`. Focusing on $k = 20$, we find that our method, consisting of `k-means` weight generation, `L2` normalizations, and `max` aggregation, outperforms all previously studied approaches by a margin of 4% on average with statistical significance. For reference, we additionally report an oracle baseline that uses cross-class feature statistics to generate its

weights, representing an upper bound that is not constrained by imprinting. The results indicate that our new imprinting method significantly narrows the gap between single-proxy `mean` imprinting and this oracle baseline.

The impact of using the unconstrained `m-nn` aggregation on `all` data is investigated at the end of this section. Next, we analyze each of the components of `IMPRINT` separately.

**Table 2 & Figure 5:** Benchmarking `GEN` mechanism for $k \leq 20$ across `FMs` and `Ts`. Best `NORM` combination for each row used implicitly. `AGG` is fixed to `max`. CD diagram proves that `k-means` weight generation is significantly better than all other methods.

| GEN | $k$ | Avg. acc. $\%$ $\pm$ std |
|---|---|---|
| k-means | 20 | **91.04** $\pm 6.26$ |
| k-medoids | 20 | 87.01 $\pm 8.23$ |
| mean | 1 | 86.84 $\pm 7.80$ |
| k-cov-max | 20 | 83.98 $\pm 9.56$ |
| k-random | 20 | 82.14 $\pm 10.15$ |
| k-fps | 20 | 65.56 $\pm 12.32$ |

**Weight Generation (`GEN`).** To assess the impact of `GEN`, we first focus on the `max` aggregation and do not fix `NORM`, but simply show the run with the best `NORM` combination, if not otherwise specified. The `m-nn` aggregation and different values for `NORM` are analyzed later in this section.

Initially, we limit the number of generated proxies to $k \leq 20$. Results in fig. 5 show how `k-means`, using as many proxies as possible (in this case, 20) outperforms by 4% on average accuracy compared to all the other `GEN` methods. The CD diagram illustrates its statistical significance in ranking. Furthermore, while `k-medoids` with 20 proxies (which necessarily have to be among the given samples, see section 3.1) is computationally expensive, it is on par with `mean`, and covariance maximization, furthest-point sampling and random selection show even weaker performances. We find similar results for $k \leq 5$, where `k-means` outperforms the other methods as well (see fig. A.1).

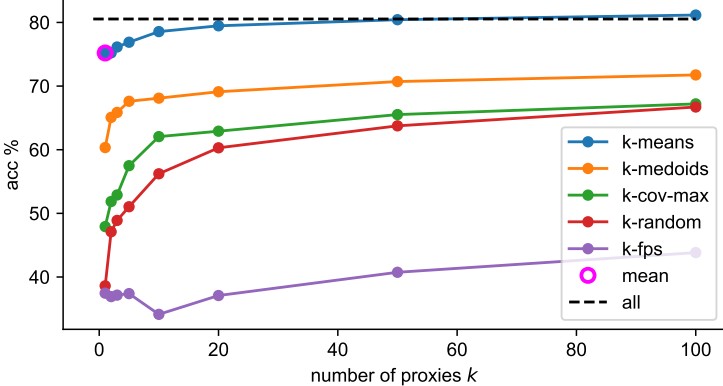

**Figure 6:** Benchmarking different `GEN` methods with `resnet18` on all of *CIFAR-10* shows superiority of `k-means` proxies. All combinations employ `L2` for all `NORM` and `max` as `AGG`.

As the number of proxies $k$ increases, `k-means` continues to be the best `GEN` method. An example for `resnet18` and *CIFAR-10* can be found in fig. 6. All methods converge towards the point of imprinting (saving) `all` data ($k = $ all), even surpassing it in the case of `k-means`. Due to its superior performance, we mainly focus on `k-means` in the remainder of the analysis.

**Learned Weights.** Employing gradient-based methods – such as setting weights via the analytical least-squares initialization proposed by Harun & Kanan (2025), which uses data *and labels* from all classes *jointly*

– can be seen as an upper bound ("Oracle" in fig. 2) for the unsupervised weight imprinting approaches discussed here. We extend this method by combining it with `k-means`, forming `k-least-squares`, and observe in appendix A.5 that using multiple proxies per class instead of a single weight vector can still improve performance, particularly on datasets with high $\mathcal{NC}_1$.

**Normalization (`NORM`).** We compare all the different `NORM` methods, focusing on `k-means` as `GEN`. For $k = 1$ and varying $\text{NORM}_{post}$ (while taking best values for $\text{NORM}_{pre}$ and $\text{NORM}_{inf}$ implicitly), fig. 7 shows that `L2` is the best choice for weight normalization. `quantile` and `none` normalization both perform worse.

**Table 3 & Figure 7:** Benchmarking $\text{NORM}_{post}$ mechanism across FMs and Ts. The best $\text{NORM}_{pre}$ and $\text{NORM}_{inf}$ combinations for each row are used implicitly. `GEN` is fixed to `mean` (that is, $k = 1$) and `AGG` is fixed to `max`. The CD diagram shows the statistical significance of `L2` as the best weight normalization $\text{NORM}_{post}$.

| $\text{NORM}_{post}$ | Avg. acc. $\%$ ± std |
|---|---|
| L2 | **86.84** ±7.80 |
| quantile | 82.90 ±12.87 |
| none | 83.26 ±9.18 |

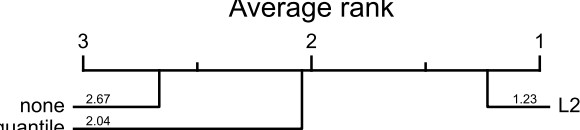

**Table 4:** Benchmarking $\text{NORM}_{pre}$ and $\text{NORM}_{inf}$ mechanisms across FMs and Ts. $\text{NORM}_{post}$ is fixed to `L2`, `GEN` to `mean`, and `AGG` to `max`. No statistically significant differences were found.

| $\text{NORM}_{pre}$ | $\text{NORM}_{inf}$ | Avg. acc. $\%$ ± std |
|---|---|---|
| none | L2 | 86.84 ±7.80 |
| none | none | 86.84 ±7.80 |
| L2 | L2 | 86.79 ±7.83 |

Keeping `L2` for $\text{NORM}_{post}$ fixed, we find no statistical differences between the different combinations of $\text{NORM}_{pre}$ and $\text{NORM}_{inf}$ (see table 4). For larger values of $k$, the differences among $\text{NORM}_{post}$ become even more pronounced. However, the performance of $\text{NORM}_{pre}$ and $\text{NORM}_{inf}$ remains statistically indifferent for `L2` weight normalization (see fig. A.2 for all combinations at once with $k = 1$, and fig. A.3 for $k = 20$).

We restrict all the subsequent experiments to using `L2` across all `NORM` following Qi et al. (2018). This combination of normalizations is chosen to specifically model cosine similarity within `max` aggregation.

**Table 5 & Figure 8:** Benchmarking `AGG` mechanism across FMs and Ts. `GEN` is fixed to `all` ($k = $ all), that is, imprinting (saving) all data to weights. `L2` normalization is used for all `NORM`. The CD diagram shows statistical significance of `3-nn`, `5-nn`, and `20-nn` over `max` aggregation.

| AGG | Avg. acc. $\%$ ± std |
|---|---|
| 5-nn | **93.74** ±4.97 |
| 3-nn | 93.50 ±5.22 |
| 20-nn | 93.56 ±4.93 |
| 1-nn | 92.81 ±5.69 |
| max | 92.81 ±5.69 |
| 50-nn | 92.91 ±5.25 |

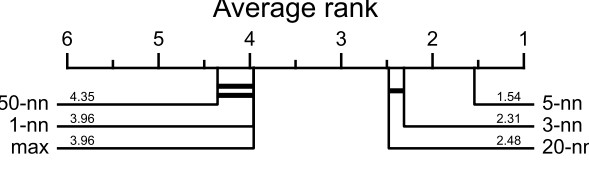

**Aggregation (`AGG`).** In addition to `max`, we study the effect of using $m$-nearest neighbor (`m-nn`) as an aggregation method. Recall that `max` is a special case of `m-nn` when $m = 1$ (as $\text{NORM}_{post}$ is set to `L2`). We investigate different values for $m \in \{1, 3, 5, 20, 50\}$.

When `all` data is imprinted, fig. 8 shows that using `m-nn` aggregation for $m \in \{3, 5, 20\}$ is slightly better than `max`. With $k = 20$ and `k-means` as `GEN`, `max` (=`1-nn`) aggregation becomes the top performing combination

**Table 6 & Figure 9:** Benchmarking `AGG` mechanism across `FMs` and `Ts`. `GEN` is fixed to `k-means` with $k = 20$. `L2` normalization is used for all `NORM`. The CD diagram shows that `max` is the best-performing aggregation method.

| AGG | Avg. acc. $\%$ $\pm$ std |
|-----|-----|
| `1-nn` | **91.06** $\pm 6.21$ |
| `max` | **91.06** $\pm 6.21$ |
| `3-nn` | 90.59 $\pm 6.16$ |
| `5-nn` | 90.12 $\pm 6.21$ |
| `20-nn` | 87.05 $\pm 7.46$ |

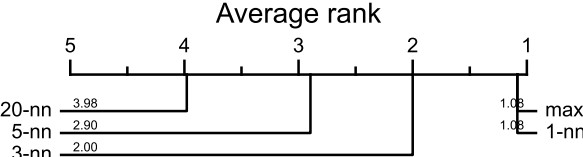

(see fig. 9). Furthermore, the reduction of proxies (from `all` ($\approx 6000$ per class) to $k = 20$) leads to a decrease in accuracy of less than $3\%$.

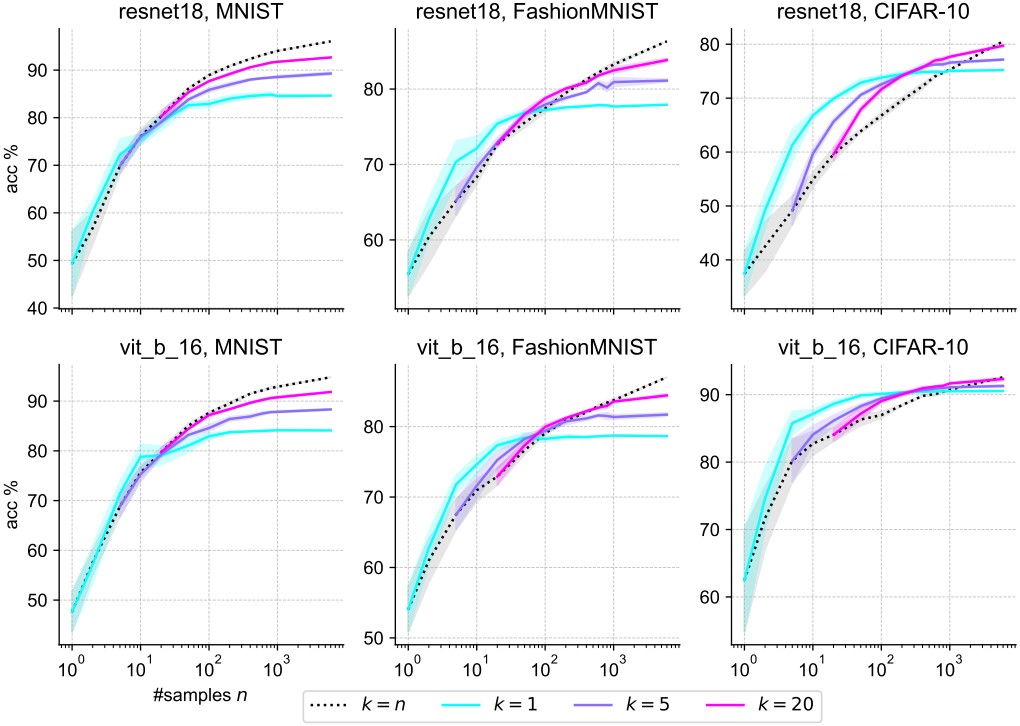

**Figure 10:** `k-means` with different values for $k$ in `n-shot` scenarios. Shaded areas indicate $95\%$ confidence intervals (CIs) over five different seeds. Other variables are fixed to our previously described best method of using `L2` normalizations and `max` as `AGG`. Note that only data for the meaningful case of $k \le n$ is shown. For *MNIST* and *FashionMNIST*, `mean` ceases to be the best strategy already at only roughly 50 samples.

## 5.2 Low-Data Regime

We analyze the `n-shot` scenario with our method (`k-means` as `GEN`, `L2` as `NORM`, and `max` as `AGG`). Furthermore, we focus on the large tasks `T` containing all ten classes of *MNIST*, *FashionMNIST*, resp. *CIFAR-10* at once. In this scenario, due to sampling only a few examples $n$, we average over five (instead of three) different seeds.

From the results in fig. 10, we find that as the number of samples $n$ increases, `k-means` starts to outperform `mean` imprinting. The use of more proxies $k$ further amplifies this performance gain. The shift occurs at roughly 50 samples per class for *MNIST* and *FashionMNIST*, while for *CIFAR-10*, $k > 1$ becomes prominently better at around 200 samples per class (see fig. A.4 for a display of all `FMs` focused on $10 \le n \le 400$).

## 5.3 Neural Collapse and Number of Proxies

Figure 11a depicts the neural collapse measurement $\mathcal{NC}_1$ (see eq. (1)) for the 100 random *ImageNet* tasks with remapped labels as explained in section 4, as well as the four tasks containing all of *MNIST*, *FashionMNIST*, *CIFAR-10*, resp. *CombiDigits*. It can be inferred that *ImageNet* has a close-to-zero $\mathcal{NC}_1$ score, which increases linearly when adding more classes to each label (i.e., increasing multi-modality). As for the other datasets, *CIFAR-10* is generally more collapsed according to its low value of $\mathcal{NC}_1$, which even falls below 1 for the Transformer-based `FMs`. We hypothesize that this is due to the similarity of its categories to those appearing in *ImageNet*. By design, the synthetic *CombiDigits* dataset, introduced in section 4, has a very high $\mathcal{NC}_1$. Apart from that, $\mathcal{NC}_1$ of the *ImageNet* data for the Transformer-based architectures are much lower and therefore they are more collapsed compared to the CNN-based `FMs`. These architectural differences are further investigated in appendix A.4.

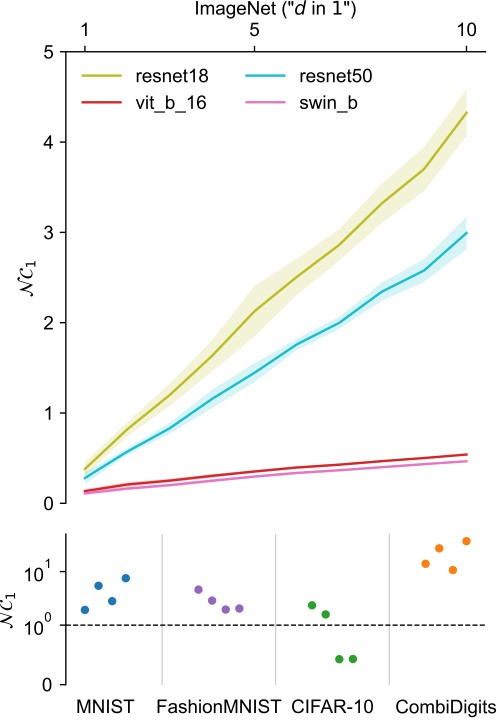

**(a)** A clear linear relationship between the neural collapse measure $\mathcal{NC}_1$ and $d$ can be inferred for all `FMs`, i.e., increased multi-modality implies less collapse. The $\mathcal{NC}_1$ of other datasets (*CombiDigits* in particular) is much higher across all `FMs`, while only `vit_b_16` and `swin_b` get an $\mathcal{NC}_1$ of less than one on *CIFAR-10*.

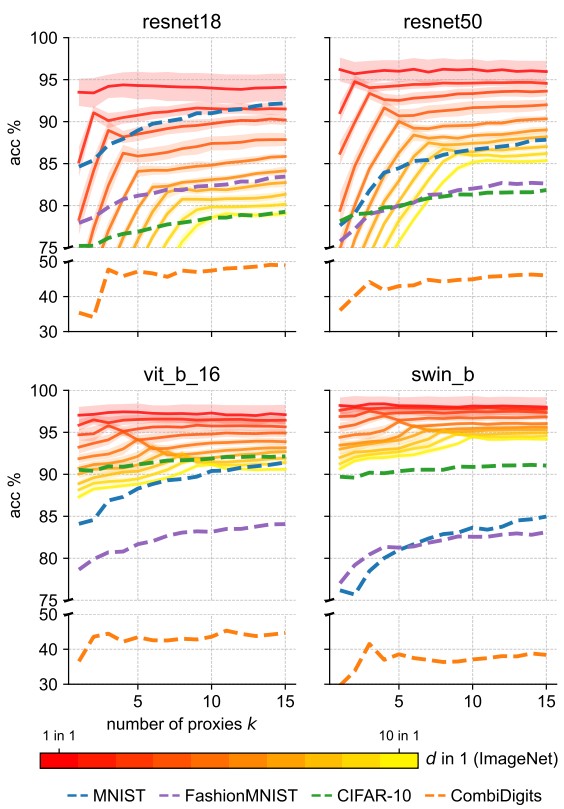

**(b)** Accuracy over number of proxies $k$ used in `k-means` together with `L2` for all `NORM` and `max` as aggregation. In all four subplots, peaks in accuracy at $k = d$ can be inferred. This confirms the connection between the effect of using multiple proxies and the collapse of the data.

**Figure 11:** Averaged $\mathcal{NC}_1$ resp. accuracy of ten random *ImageNet* label remappings ("$d$ in 1") for every $d = 1, \ldots, 10$. Shaded areas indicate 95% CIs over three different seeds. Values for the tasks containing all of *MNIST*, *FashionMNIST*, *CIFAR-10*, resp. *CombiDigits* at once are shown in dotted styles.

For the same data, fig. 11b depicts accuracy over a varying number of proxies $k$ inferred from `k-means`. A prominent peak at $k = d$ can be inferred for every `FM` and reflects that $d$ class proxies lead to the best result for $d$-modal class distributions. Furthermore, increasing $k$ for the *ImageNet* sets has a much larger effect on the CNN-based `FMs`. We argue that this is because of their higher values of $\mathcal{NC}_1$, indicating less neural collapse. In appendix A.4, we analyze architectural differences and training setups to explain these variations. The fact that *CIFAR-10* has the lowest $\mathcal{NC}_1$ scores (see fig. 11a) is reflected by flat green curves over $k$.

Figure 12 renders this connection more precise: For datasets with large $\mathcal{NC}_1$, i.e., where intra-class variability surpasses inter-class variability, using more than a single proxy per class ($k > 1$) yields a clear performance gain over mean imprinting ($k = 1$).

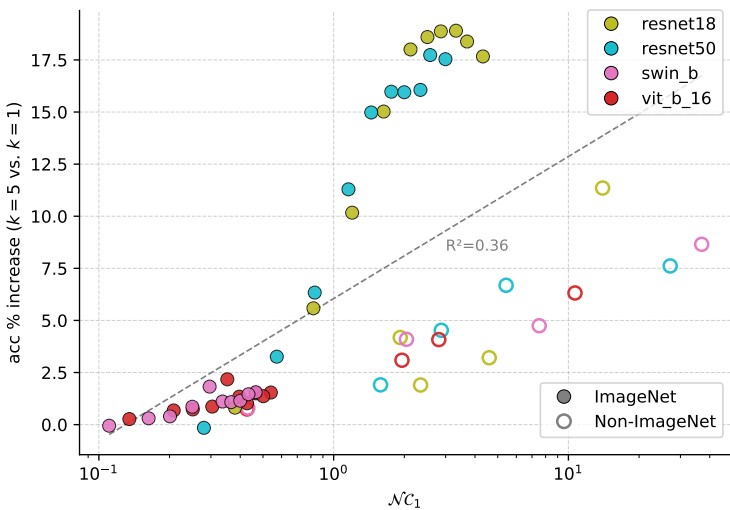

**Figure 12:** Accuracy (%) gain when using $k = 5$ instead of $k = 1$ proxies per class with `k-means`, plotted against the neural collapse measure $\mathcal{NC}_1$ (log-scaled axis). Colors indicate different `FM`s. Outlined markers correspond to non-*ImageNet* datasets (*MNIST*, *FashionMNIST*, *CIFAR-10*, and *CombiDigits*), while filled markers denote averaged results from our random *ImageNet* label remappings. The dashed gray line shows a least-squares fit ($R^2 = 0.36$), summarizing an approximate log-linear dependence between $\mathcal{NC}_1$ and the gains from using $k > 1$ proxies. Across all points, the Spearman rank correlation is $\rho = 0.82$ ($p < 0.0001$), indicating a statistically significant positive association. Consistent with this trend, for regimes with $\mathcal{NC}_1 > 1$ single-proxy `mean` imprinting ($k = 1$) appears to become substantially suboptimal compared to using multiple proxies ($k > 1$).

## 6 Conclusion

We present a new framework, `IMPRINT`, to analyze three main components relevant to weight imprinting, namely, weight generation, normalization, and aggregation. Within `IMPRINT`, state-of-the-art imprinting strategies become special cases. This allows for a comprehensive analysis of different approaches through systematic experiments and leads us to generalize to a new, best-performing imprinting variant. That is, using `k-means` weight generation with `L2` normalizations and `max` aggregation, which outperforms all previously studied methods (see fig. 2).

**k-means generates better weights than mean.** In particular, we find that the `mean` weight generation (`GEN`) method, despite its prominence in previous work, falls short compared to `k-means` – even when the number of proxies $k$ is very small. Remarkably, with as little as 50 samples per class, `k-means` can already outperform the original imprinting method proposed by Qi et al. (2018), highlighting its advantage in low-data regimes.

**L2 weight normalization is essential for strong performance.** The `max` aggregation directly scales with the magnitude of the weights. Normalization ($\texttt{NORM}_{post}$) ensures that all proxies have the same magnitude, preventing differences in vector norms from disproportionately affecting classifier predictions. Nearest neighbor (`1-nn`) aggregation is not as affected by the lack of normalization, since it uses Euclidean distance. Although still part of common procedure, normalizations for embeddings ($\texttt{NORM}_{pre}$ and $\texttt{NORM}_{inf}$) appear to have minimal impact on performance.

**With `max` aggregation, there is no need to store `all` data.** While nearest neighbor (`m-nn`) aggregation (`AGG`) performs well when all data is saved (e.g., when there are no storage constraints), `max` aggregation with limited number of representative proxies (e.g., through $k$-means) is an efficient alternative without a substantial loss in performance.

**Neural collapse proves the efficacy of imprinting.** During training, the last-layer weights of a `FM` tend to collapse to their respective class means. This proves the success of `mean` imprinting on known classes. New, out-of-distribution data, however, often shows less collapse, making it beneficial to imprint more than one proxy. Our results show that the advantage of using multiple proxies is strongly coupled to the degree of neural collapse: the accuracy gain from using $k > 1$ proxies increases approximately log-linearly with $\mathcal{NC}_1$. In particular, multi-proxy imprinting yields substantial and predictable improvements on datasets with $\mathcal{NC}_1$ exceeding 1. While from a practical perspective, choosing the exact number of proxies $k$ based on pure greedy search with validation data (or as part of any AutoML pipeline) is still a valid option, our analysis provides insights into the underlying mechanism, turning $\mathcal{NC}_1$ into an indicator for when multi-proxy imprinting is highly benefitial.

**Limitations.** Our experiments are limited to foundation models for image classification and do not cover other data modalities such as audio, text, or sensor inputs. While `IMPRINT` is agnostic to modality, empirical validation of our results outside of vision is needed. Although imprinting alone provides an efficient solution to transfer learning, when compared to purely gradient-based learning, a gap still remains. We do not investigate the benefit of combining it with optimization methods such as gradient-based learning, and the choice of $k$ still requires heuristic or empirical selection rather than direct prediction.

**Future Work.** The usage of both weight and activation sparsity as Shen et al. (2023) could change the within- and between-class variability in favor of using a higher number of proxies. Besides only using the penultimate layer embeddings for generating the classifier weights, an interesting area of study could be extracting embeddings from previous layers of the `FM` for this purpose. Recently, the study by Marczak et al. (2025) showed that adding a multi-layer perceptron projector between the penultimate and classification layers results in representations that are more transferable. Another avenue of research is to thoroughly analyze imprinting the weights of other layers as well (Siam et al., 2019).

### Acknowledgments

Our work is funded by the Deutsche Forschungsgemeinschaft (DFG, German Research Foundation) – FIP-12 – Project-ID 528483508, as well as the European Union under the grant project 101079894 (COMFORT - Improving Urologic Cancer Care with Artificial Intelligence Solutions). Views and opinions expressed are however those of the author(s) only and do not necessarily reflect those of the European Union or European Health and Digital Executive Agency (HADEA). Neither the European Union nor the granting authority can be held responsible for them. We thank Viet Anh Khoa Tran for initial discussions about the neural collapse phenomenon. We further thank the reviewers and the Action Editor for their detailed and constructive feedback, which substantially improved this work.

### Author Contributions

JW contributed to the development of the framework, conducting experiments and evaluated the findings. GA was responsible for investigating NC measures and overall contribution to the project. MK contributed to extending the framework and handling data preparation. AF provided critical feedback on the presentation of the results and contributed to refining the manuscript. AL, ER, and FG provided supervision, contributed to the overall concepts presented, and to refining the manuscript.

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

# A   Appendix

## A.1   Additional Results

We provide additional tables and critical difference (CD) diagrams that expand on the results of section 5.

**Table A.1 & Figure A.1:** Benchmarking `GEN` mechanism for $k \leq 5$ across `FMs` and `Ts`. Best `NORM` combination for each row is used implicitly. `AGG` is fixed to `max`. CD diagram depicts statistical significance of `k-means` as `GEN`. See fig. 5 for $k \leq 20$.

| GEN | $k$ | Avg. acc. % ± std |
|---|---|---|
| k-means | 5 | **89.15** ±6.96 |
| mean | 1 | 86.84 ±7.80 |
| k-medoids | 5 | 84.87 ±8.76 |
| k-cov-max | 5 | 82.11 ±10.39 |
| k-random | 5 | 75.93 ±11.92 |
| k-fps | 5 | 63.64 ±12.29 |

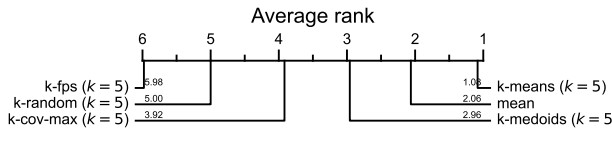

**Table A.2 & Figure A.2:** Benchmarking `NORM` across `FMs` and `Ts` shows crucial effect of L2 normalization. `GEN` is fixed to `mean` and `AGG` to `max`. CD diagram depicting statistical significance of `L2` for $\text{NORM}_{post}$. Combinations are listed as "$\text{NORM}_{inf}$ & $\text{NORM}_{pre}$ & $\text{NORM}_{post}$".

| $\text{NORM}_{inf}$ | $\text{NORM}_{pre}$ | $\text{NORM}_{post}$ | Avg. acc. % ± std |
|---|---|---|---|
| L2 | none | L2 | **86.84** ±7.80 |
| none | none | L2 | **86.84** ±7.80 |
| L2 | L2 | L2 | **86.79** ±7.83 |
| L2 | none | quantile | 82.90 ±12.87 |
| none | none | quantile | 82.90 ±12.87 |
| L2 | L2 | quantile | 82.83 ±12.86 |
| L2 | L2 | none | 83.26 ±9.18 |
| L2 | none | none | 67.66 ±22.12 |
| none | none | none | 67.66 ±22.12 |

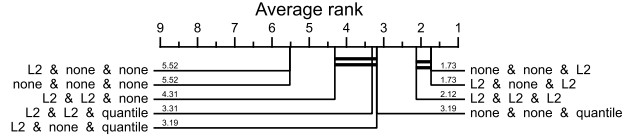

**Table A.3 & Figure A.3:** Benchmarking `NORM` across `FMs` and `Ts`. `GEN` is fixed to `k-means` with $k = 20$ and `AGG` to `max`. CD diagram depicting statistical significance of `L2` for $\text{NORM}_{post}$. Combinations are listed as "$\text{NORM}_{inf}$ & $\text{NORM}_{pre}$ & $\text{NORM}_{post}$."

| $\text{NORM}_{inf}$ | $\text{NORM}_{pre}$ | $\text{NORM}_{post}$ | Avg. acc. % ± std |
|---|---|---|---|
| L2 | none | L2 | **91.04** ±6.26 |
| none | none | L2 | **91.04** ±6.26 |
| L2 | L2 | L2 | **91.06** ±6.21 |
| L2 | L2 | quantile | 90.51 ±6.40 |
| L2 | L2 | none | 89.55 ±6.70 |
| L2 | none | quantile | 79.21 ±15.20 |
| none | none | quantile | 79.21 ±15.20 |
| L2 | none | none | 73.53 ±21.05 |
| none | none | none | 73.53 ±21.05 |

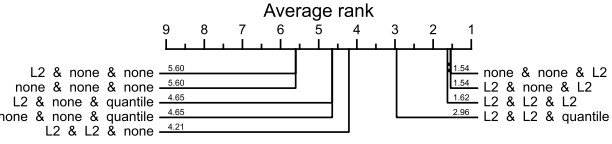

## A.2   Datasets

We briefly describe the datasets used in our experiments.

***ImageNet* (Deng et al., 2009).**   We use the ILSVRC 2012 version (commonly called *ImageNet*-1K) containing 1 000 classes and 1.2M training images. Since the test set is not publicly available, we use the

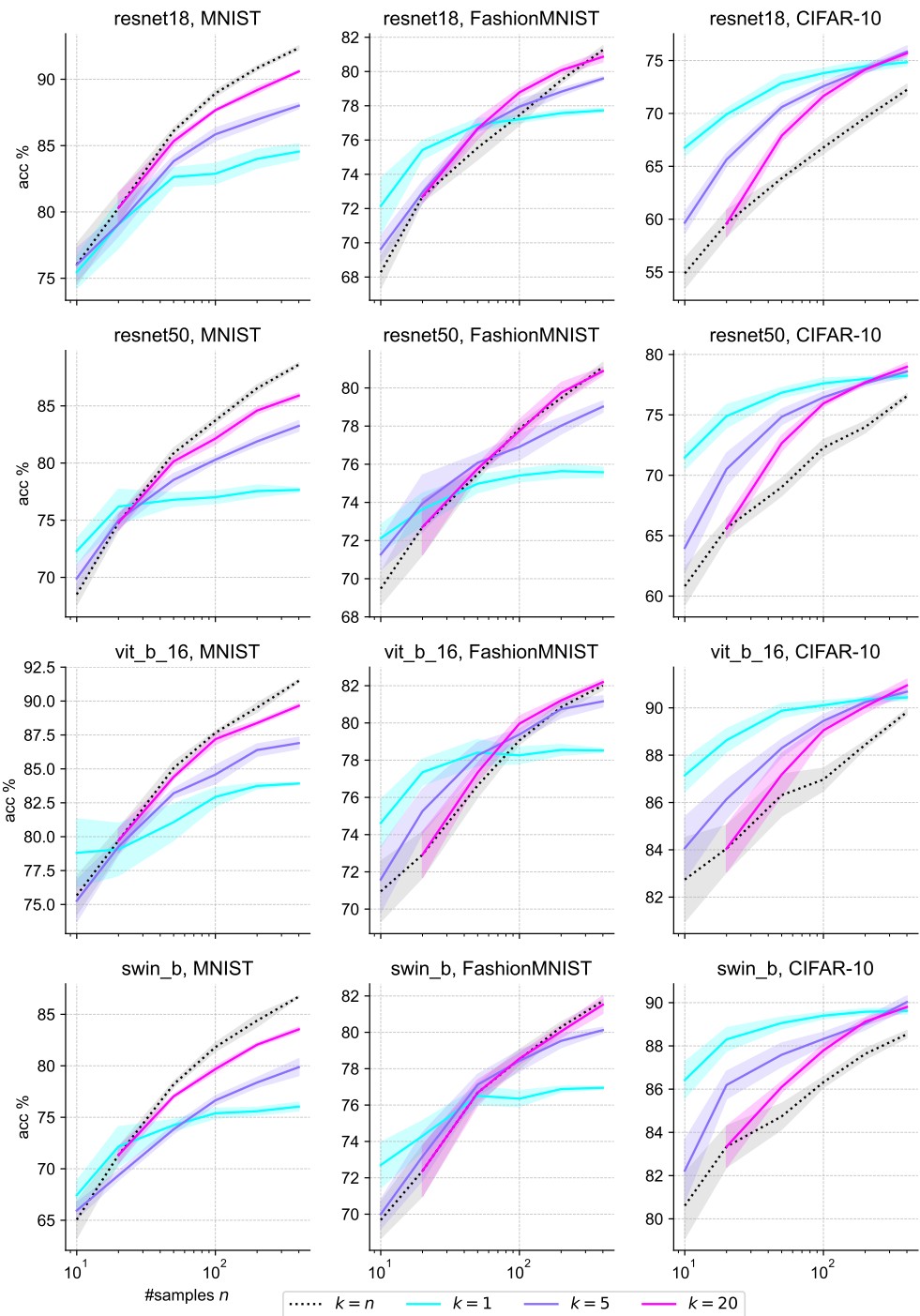

**Figure A.4:** k-means with different values for $k$ in n-shot scenarios with focus on $10 \leq n \leq 400$. Shaded areas indicate 95% confidence intervals (CIs) over five different seeds. Other variables are fixed to our previously described best method of using L2 for NORM and max as AGG. Note that only data for the meaningful case of $k \leq n$ is shown. For *MNIST* and *FashionMNIST*, mean ceases to be the best strategy already at only roughly 50 samples. See fig. 10 for more values of $n$.

validation set (50 000 images) as a stand-in. For neural collapse investigations, we construct tasks by randomly grouping $d$ classes into one label, producing "$d$ in 1" tasks as explained in section 4.

***MNIST* (Deng, 2012).** A benchmark dataset of handwritten digits (0–9), consisting of 60 000 training and 10 000 test grayscale images of size $28 \times 28$.

***FashionMNIST* (Xiao et al., 2017).** A drop-in replacement for *MNIST* with the same format and number of samples, containing grayscale images of fashion items across 10 classes.

***CIFAR-10* (Krizhevsky et al., 2009).** A dataset of $32 \times 32$ RGB images covering 10 object classes, with 50 000 training and 10 000 test samples.

***MNIST-M* (Ganin et al., 2016), *SVHN* (Netzer et al., 2011), *USPS* (Hull, 1994).** Digit classification datasets each containing digits 0-9 with domain-specific visual characteristics. *MNIST-M* applies color and texture transformations to MNIST digits, yielding 60 000 training and 10 000 RGB images of size $28 \times 28$. *SVHN* consists of digit crops from house numbers in Google Street View, totaling 73 257 training and 26 032 test images of $32 \times 32$ in RGB. *USPS* contains scanned and normalized handwritten digits in $16 \times 16$ grayscale, split into 7 219 training and 2 007 test images.

***CombiDigits* (Ours).** A synthetic dataset constructed by merging all classes from *MNIST*, *MNIST-M*, *SVHN*, and *USPS*. Each class label corresponds to a digit (0–9) and fig. A.5 shows example images of class "6". The dataset includes significant visual heterogeneity across sources and thus simulates multi-modal, non-collapsed class distributions.

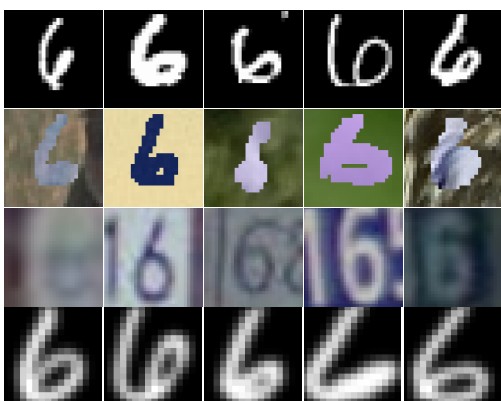

**Figure A.5:** Twenty sample images from class "6" of the *CombiDigits* dataset. Each row corresponds to one of the four source datasets: *MNIST*, *MNIST-M*, *SVHN*, and *USPS* (top to bottom), illustrating the intra-class heterogeneity across domains.

### A.3 Imprinting as Memory

We revisit the classical idea of Bidirectional Associative Memories (BAMs) (Kohonen, 2009; Anderson, 1972; Nakano, 2007; Anderson et al., 1977) and the associated update rule (Sejnowski, 1977; Dayan & Willshaw, 1991) for storing key-value pairs as in

$$\mathbf{W} \leftarrow \mathbf{W} + \mathbf{v}\mathbf{k}^\top,$$

where $\mathbf{W} \in \mathbb{R}^{l \times l}$ is a matrix and $\mathbf{k}, \mathbf{v} \in \mathbb{R}^d$ are key and value vectors, respectively, to be stored therein.

Throughout this work, imprinting can be interpreted as inserting such key-value associations into the classifier weights. More precisely, the generation (GEN) component extends the linear classification head by setting $\mathbf{v}$

as a one-hot vector representing the class, and $\mathbf{k}$ as the corresponding proxy previously computed. While we focus on this simple linear setting, imprinting is not limited to it, as discussed in section 2.

Notably, this form of direct memory update has seen renewed attention in modern architectures beyond standard query-key-value attention. In particular, the xLSTM model (Beck et al., 2024) implements this mechanism within its mLSTM memory blocks, where the matrix memory cell is updated by gated key-value associations, closely following this classical covariance rule, indicating a broader resurgence of associative memory principles in contemporary sequence modeling.

### A.4 Differences between Foundation Models

While an in-depth comparison of foundation models is beyond the scope of this work, we believe it is important to highlight key observations we have made. In particular, fig. 11a shows significantly lower $\mathcal{NC}_1$ scores for `vit_b_16` and `swin_b` on their pre-training *ImageNet* data compared to the `resnet` models. We hypothesize that this difference is primarily due to model size and training regimes. The Transformer-based architectures (`vit_b_16` and `swin_b`) have a considerably higher parameter count ($\approx 87M$) than the `resnet` models (11.7M and 25.6M, respectively). Additionally, `vit_b_16` and `swin_b` were trained for more than three times as many epochs (300 vs. 90) while using a substantially lower learning rate (0.003 and 0.01 vs. 0.1). Notably, the embedding dimensions of these models are comparable, meaning that the observed differences in $\mathcal{NC}_1$ scores cannot be attributed to differences in representation dimensionality. Instead, we argue that the combination of larger model size, extended training duration, and lower learning rates likely contributes to greater overfitting, leading to more pronounced collapse. As discussed in section 5.3, this enables the Transformer-based `FM`s to handle the *ImageNet* tasks with remapped labels more effectively and to achieve significantly better performance on the similarly distributed *CIFAR-10* dataset.

Figure A.6a, similar to fig. 11b, illustrates the impact of varying the number of proxies on imprinting accuracy across different foundation models (`FM`s). The key difference in this figure is the use of `none` for $\text{NORM}_{pre}$ instead of `L2`. This seemingly minor change reveals a striking contrast between CNN- and Transformer-based architectures: a distinct and consistent dip between $k = 1$ and $k = d$ appears in Transformer-based models, whereas this dip is absent in fig. 11b, where `L2` is used as $\text{NORM}_{pre}$, and does not occur at all in the `resnet` models. We hypothesize that this difference arises from the distinct embedding distributions of CNN- and Transformer-based architectures (see, e.g., Hosoda et al. (2024, Figure S2)).

### A.5 Learned Weights (with Multiple Proxies) and Comparisons

Harun & Kanan (2025) recently studied the initialization of classifier weights for novel categories in a Continual Learning (CL) setting using the least squares algorithm, comparing it to random initialization and class mean imprinting across various loss functions. In contrast, our work centers on imprinting-based approaches, which avoid using gradient-based optimization and cross-class statistics, operating instead on a per-class basis with immediate availability of data. Nonetheless, we include the least-squares method as a supervised oracle baseline – explicitly not an imprinting method – fine-tuned for classification accuracy.

We evaluate the performance of `least-squares` on the same tasks as defined in section 4. Additionally, we introduce a multi-proxy extension, `k-least-squares`, which combines least squares with `k-means` clustering. As in the case of weight generation `GEN` in imprinting (where `k-means` outperforms `mean`), we find that using multiple proxies per class improves performance in settings with high $\mathcal{NC}_1$ as well, indicating that the benefits of multi-prototype representations persist even in this non-imprinting, supervised context.

All experiments use no normalization (`none` as `NORM`), based on ablations confirming that additional normalization impairs performance. This matches expectations, since least squares outputs are already calibrated, and normalization distorts them.

**Least Squares Weights.** For all data contained in task `T`, define $\mathbf{H} \in \mathbb{R}^{l \times N}$ as the collection of all feature vectors of the $N$ training samples in `T` obtained by applying a fixed `FM`, i.e., $\mathbf{h}_{c,i} \in \mathbb{R}^l$ is the feature vector of the $i$-th sample in the $c$-th class. Recall from section 3.2 that the within-class covariance matrix $\mathbf{\Sigma}_W$ is given as $\mathbb{E}_{c,i}[(\mathbf{h}_{c,i} - \overline{\mathbf{h}}_c)(\mathbf{h}_{c,i} - \overline{\mathbf{h}}_c)^\top]$ and additionally define the total covariance matrix $\mathbf{\Sigma}_T$ and the class-means

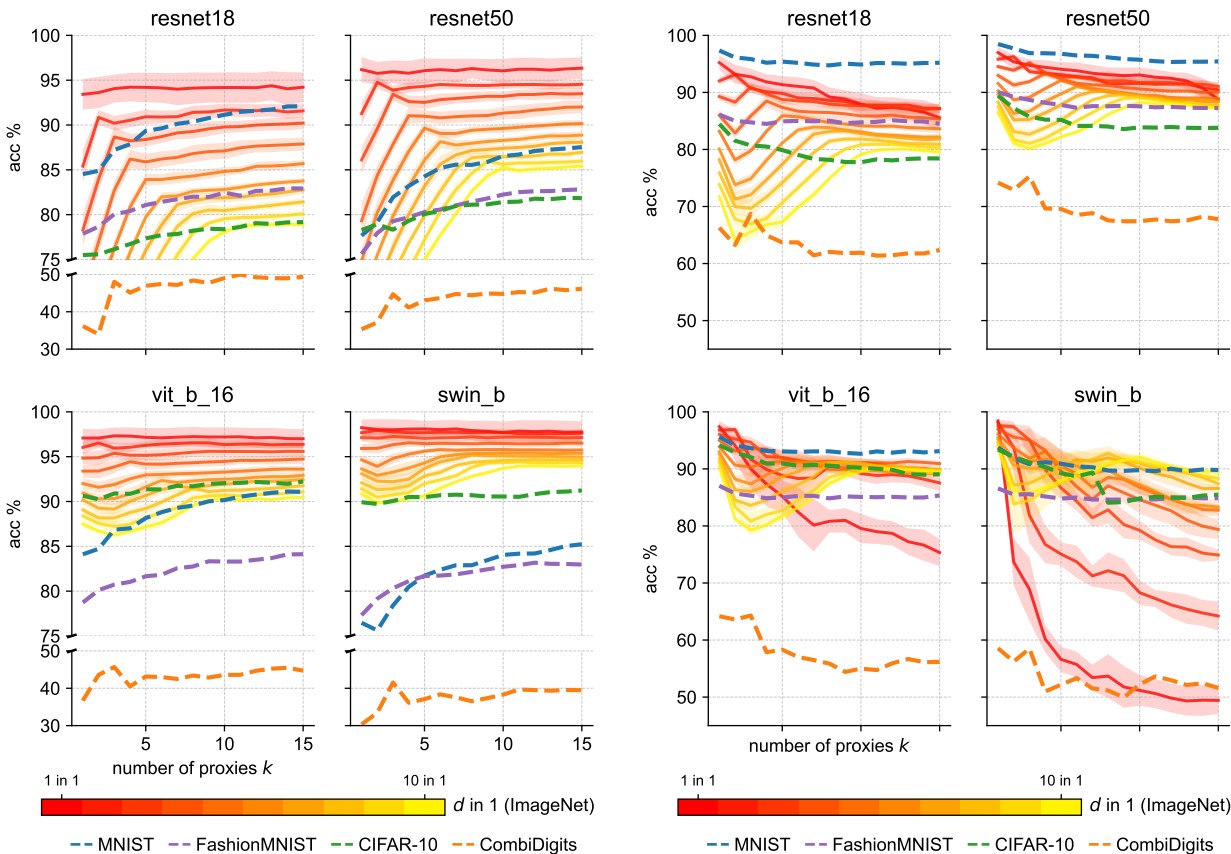

**(a)** `k-means` used as `GEN`. $\text{NORM}_{post}$ and $\text{NORM}_{inf}$ are set to `L2`, and $\text{NORM}_{pre}$ to `none`. Besides the prominent peaks in accuracy at $k = d$ (as already observed in fig. 11b), a consistent dip between $k = 1$ and $k = d$ appears in Transformer-based models, which was not to be seen with $\text{NORM}_{pre}$ set to `L2` as well. In appendix A.4 we hypothesize that this is due to the distinct embeddings distributions of CNN- and Transformer-based architectures.

**(b)** `k-least-squares` (see appendix A.5) used as `GEN` and all `NORM` are set to `none`. Here, the CNN-based `FM`s exhibit a clear accuracy peak at $k = d$, with a consistent drop for intermediate values $1 < k < d$. For the Transformer-based `FM`s, however, accuracy is typically highest at $k = 1$ and only improves slightly around $k = d$ but does not show a pronounced peak. For all `FM`s, performance declines with $k > d$, indicating that excessive proxy splitting harms generalization. This degradation does not occur in `k-means` `GEN`, which remains robust even with large $k$ (see figs. 11b and A.6a).

**Figure A.6:** Averaged accuracy of ten random *ImageNet* label remappings ("*d* in 1") for every $d = 1, \ldots, 10$ over number of proxies $k$. Shaded areas indicate 95% CIs over three different seeds. Values for the tasks containing all of *MNIST*, *FashionMNIST*, *CIFAR-10*, resp. *CombiDigits* at once are shown in dotted lines.

matrix $\mathbf{M}$ as

$$\mathbf{M} = [\overline{\mathbf{h}}_1, \ldots, \overline{\mathbf{h}}_C] \in \mathbb{R}^{l \times C}, \qquad \mathbf{\Sigma}_T = \mathbb{E}_{c,i}[(\mathbf{h}_{c,i} - \mathbf{h}_G)(\mathbf{h}_{c,i} - \mathbf{h}_G)^\top] \in \mathbb{R}^{l \times l}.$$

From these feature statistics, we can obtain the `least-squares` weights via

$$\mathbf{W}_{LS} = \frac{1}{C}\mathbf{M}^\top (\mathbf{\Sigma}_T + \mathbf{h}_G \mathbf{h}_G^\top + \lambda \mathbf{I})^{-1}. \tag{A.1}$$

Here, $\mathbf{I}$ is the identity matrix and $\lambda$ is the weight decay. We set $\lambda$ to match the value used during the original training of each respective model. Table A.4 lists the specific $\lambda$ values used for all models considered in our experiments.

**Table A.4:** Weight decay values $\lambda$ used for each model.

| Model | $\lambda$ |
|---|---|
| `resnet18`, `resnet50` (He et al., 2016) | 0.0001 |
| `vit_b_16` (Dosovitskiy et al., 2021) | 0.1 |
| `swin_b` (Liu et al., 2021) | 0.05 |

Equation (A.1) reveals that `least-squares` shares structural similarities with `mean` imprinting, but applies a more sophisticated normalization scheme that includes both scaling and rotation. Crucially, it relies on cross-class statistics computed across the entire dataset, and thus is incompatible with imprinting scenarios, which operate by directly constructing class weights from the data of a single class without supervision or access to other classes.

---

**Algorithm 1** `k-least-squares`

---

**Input:** Class data $\{\mathbf{H}_c\}_{c=1}^C$, number of proxies $k$
**Output:** Weights $\{\mathbf{W}_c\}_{c=1}^C$ with shape $[k, l]$ per class
 1: **if** $k = 1$ **then**
 2:     **return** standard least squares weights $\mathbf{W}_{LS}$ for each class (see eq. (A.1))
 3: **end if**
 4: **for** each class $c$ **do**
 5:     Cluster $\mathbf{H}_c$ into $k$ clusters via $k$-means
 6:     Assign each cluster $j$ to proxy class $(c, j)$
 7:     Collect proxy samples $\{\mathbf{H}_{(c,j)}\}_{j=1}^k$
 8: **end for**
 9: Compute least squares weights $\mathbf{w}_{(c,j)}$ jointly for all proxy classes $(c, j)$
10: Assemble $\mathbf{W}_c = [\mathbf{w}_{(c,1)}, \ldots, \mathbf{w}_{(c,k)}]$ for each original class $c$
11: **return** $\{\mathbf{W}_c\}_{c=1}^C$

---

**Combining `k-means` and `least-squares` into `k-least-squares`.** To allow for multiple proxies per class, we propose `k-least-squares`, a generalization of standard least squares (`least-squares`) that integrates clustering. Instead of computing a single weight vector per class using all class samples, we first partition each class's feature set $\mathbf{H}_c$ into $k$ clusters using $k$-means. Each cluster is treated as a separate proxy class, effectively expanding the classification task from $C$ to $k \cdot C$ proxy classes. We then solve a single regularized least squares problem over all proxy classes at once, assigning a distinct target vector to each proxy. The resulting weights are grouped by their original class to yield $k$ weight vectors per class. The complete procedure is given in algorithm 1.

The results in fig. A.6b extend previous findings from `mean` and `k-means` for the CNN-based `FM`s, as accuracy increases with increasing proxy count $k$, peaks at $k = d$, and reveals a distinct dip between $k = 1$ and $k = d$, confirming the weakness of using only one proxy per class. For Transformer-based `FM`s, by contrast, accuracy is often highest at $k = 1$, improves slightly near $k = d$, but lacks a pronounced peak. Beyond $k > d$, accuracy generally declines for `k-least-squares`, indicating diminishing returns from excessive proxy splitting. In contrast, `k-means GEN` maintains stable performance even as $k$ grows, demonstrating robustness to increasing proxy counts.

**Comparison with `k-means`.** We now compare the performance of `k-means` and `k-least-squares` as proxy generation methods, noting again that `k-least-squares` is not an imprinting scheme as it does not operate on a class-by-class basis as the imprinting methods presented in section 3.1 do.

Upon proper comparison of the best performing `k-means` and `k-least-squares` configurations, as depicted in fig. A.7, we observe that on our synthetic *ImageNet* tasks, `k-least-squares` does not consistently offer a substantial improvement. In fact, it is only on `resnet50`, in a highly multi-modal setting ($d = 10$), that

`k-least-squares` reaches better accuracy. For the Transformer-based models, `k-means` even performs better in these scenarios.

While the Transformer-based models generally achieve superior performance overall (note the different y-axis scales for CNN- and Transformer-based models in fig. A.7), a consistent trend across all models is observed: with increasing multi-modality (increasing $d$), using more than one proxy, whether through `k-means` or `k-least-squares`, begins to outperform the single-proxy methods (`mean` and `least-squares`). Still, it is striking to see how effective single weights with `least-squares` can be. However, it is also noteworthy how easily `k-means` can bridge this gap, particularly considering that `least-squares` is not an actual imprinting scheme, while `k-means` is.

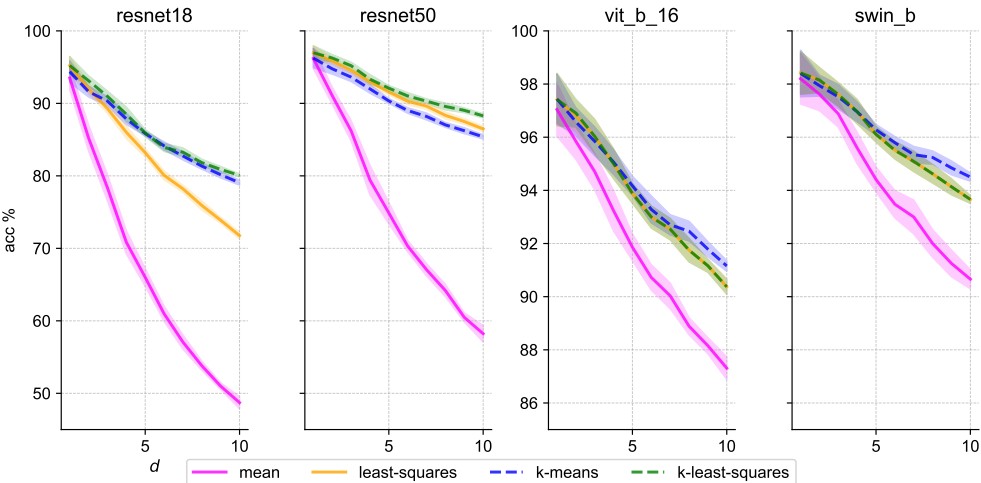

**Figure A.7:** Averaged accuracy of ten random *ImageNet* label remappings ("$d$ in 1") for every $d = 1, \ldots, 10$ for `mean` and `least-squares`, and optimal `k-means` and `k-least-squares` ($k \in \{0, \ldots, 15\}$) weights. Shaded areas indicate 95% CIs over three different seeds. Note the differing y-axis scales for CNN-based (`resnet18`, `resnet50`) and Transformer-based (`vit_b_16`, `swin_b`) models. The figure illustrates that with increasing multi-modality $d$, multi-proxy methods (`k-means`, `k-least-squares`) generally outperform single-proxy methods (`mean`, `least-squares`), and that the `k-means` imprinting scheme is competitive with `least-squares` approaches.

Furthermore, we evaluate the `k-least-squares` weight generation method (using `none` for all `NORM`) across the 48 tasks proposed in the main part of our paper (see section 4). `least-squares` reaches an average accuracy of 94.54%, while `20-least-squares` **drops** to 91.20%. The observed decrease in accuracy can be explained by the dynamics shown in fig. A.6b. In comparison to the numbers presented in fig. 2, this shows that our $k$-means imprinting scheme significantly bridges the gap between single-proxy imprinting ($k = 1$) and the optimal least squares weights.

## A.6 Computational Efficiency

**Clustering-based imprinting.** Our `k-means` implementation in `GEN` uses `sklearn.cluster.KMeans` with its default parameters. The cost is $\mathcal{O}(Nklt)$ for $N$ samples, $k$ clusters, feature dimension $l$, and $t$ Lloyd iterations. While $k$-means is not computationally negligible, both assignment and update steps in each iteration parallelize naturally, and convergence typically requires only a few iterations. Similarly, the other steps in our imprinting framework (`NORM`, `AGG`) scale linearly with dataset size or proxy count and parallelize efficiently, resulting in no practical scalability bottlenecks even for larger or more complex datasets.

**Gradient-based optimization.** The closed-form `least-squares` costs $\mathcal{O}(Nl^2 + l^3 + NlC)$ from the covariance computation and matrix inversion. If solved through stochastic gradient descent, the cost is $\mathcal{O}(NlCt)$ with $t$ epochs.

**Empirical runtime.** Table A.5 shows that `least-squares` is faster than `k-means` in practice, likely because it processes all data in a single closed-form step rather than iterating class-by-class. Nevertheless, it should be clarified again that `least-squares` represents the analytic, non-iterative optimum derived from gradient minimization and lacks sequential class handling – precisely where imprinting excels (e.g., in continual or edge-learning scenarios).

**Table A.5:** Average runtime in seconds for different `GEN` variants across all 48 tasks (see section 4). To ensure transparency, all timing measurements were obtained on identical hardware (8 vCPUs on Intel Xeon Gold 6438Y+ nodes (2 sockets, 64 physical cores/128 threads)). Cores were not pinned to processes, which may introduce minor variance; however, we observed less than 10% variation across seeds.

| `GEN` variant | Runtime (s) |
|---|---|
| `mean` | 0.0029 |
| `5-means` | 1.2249 |
| `20-means` | 1.7321 |
| `least-squares` | 0.1593 |
| `5-least-squares` | 1.3724 |
| `20-least-squares` | 2.3717 |

