# OpenReview forum: "Robust Weight Imprinting: Insights from Neural Collapse and Proxy-Based Aggregation"
_TMLR — Accepted by TMLR_

### Review · Reviewer_n2pB · 2025-08-10

**Summary Of Contributions:**

This paper proposed the Imprint framework to address the transfer learning for unseen classes. The framework consists of three steps: GEN, NORM, and AGG, which are based on the embeddings of pre-trained models. It does not require any cross-class statistics and gradient optimization. The details of the three steps are sufficiently explained, and the experiment demonstrates that the K-means GEN and L2 normalization NORM, along with the appropriate k-NN AGG, can provide state-of-the-art results. Also, the comprehensive analyses were conducted, which show that the proposed algorithm can avoid the Neural Collapse problems.

**Additional Comments:**

Minor comments:

k-cov-max:

If you can provide a simple description for this algorithm, the reader can gain a more comprehensive understanding.

effect of $d$:

In the neural collapse, the effect of $d$ is interesting. Can you provide a more insightful explanation, and provide a criterion?

**Audience:**

Yes

**Audience Explanation:**

The algorithm is beneficial in many areas, and the representational power of deep networks can be analyzed well. It can enrich the study concerning the representative learning.

**Broader Impact Concerns:**

There are no critical ethical issues. Also, the study can be used in the more elaborate transfer learning, which can have practical ways to various domains and industries.

**Claims And Evidence:**

Yes

**Claims Explanation:**

There are ablation results to be used for the validation of the claim, especially for the neural collapse probelms. The disadvantages are that 1) there is no comparison with the gradient optimization algorithm, 2) It is not clear why the best option in GEN is k-means and others.

**Requested Changes:**

First, the more intensive experiments compared to gradient-based or other type of algorithms. The process to find the optimal hyperparameters is well explained and sufficient. However, the comparison over various types is limited.
Second, more analytic results can be required. Especially, the difference between k-means and k-medoids should be highlighted, as their performances are significantly different.

---

### Review · Reviewer_81QZ · 2025-08-28

**Summary Of Contributions:**

The paper examines "imprinting", a branch of transfer learning that avoids parameter optimization (even for the classification head). Imprinting works in two stages. To train, we obtain representative "proxy" (or "prototype") vectors for each class by aggregating embeddings of a training set. To make predictions, a test image's embedding is compared to each class' proxy, and a nearest neighbor (or m-nearest neighbors) approach converts these distances to a predicted class.

This paper suggests a general framework for imprinting, focusing on 3 key design decisions: generation of proxies (at training), normalization of embeddings (at training and test time), and aggregation (at test time to make a prediction). The recommended pipeline involves k-means to obtain proxies, L2 normalization, and m-nearest neighbor aggregation when possible.

The paper also makes a connection to neural collapse. A previous quantitative measure of collapse (variability collapse, see Eq 1), which gives smaller scores when within-class (co)variance is smaller and when between-class (co)-variance is larger, is said to coorelate with the need for many proxies.

Experiments on several image datasets evaluate lots of different instances of the IMPRINT framework, identifying the GEN, AGG steps that give on average the best performance.

**Additional Comments:**

## Minor: poor phrasing: "all weights contribute equally to the output"

I'm not a fan of this phrase, found in Conclusion. Maybe instead, you want to say that by ensuring the magnitude of each class-proxy vector is the same, you avoid problems where differences in vector magnitudes make big differences in classifier predictions.


## Minor: how many examples is "few shot"?

My reading of "few shot" literature is that they often focus on just 5 or 10 examples per class. The present paper suggests (e.g. bottom of page 11) that 50 or 200 examples is where the existing framework really shines... but is that "few shot"? I'm open to a big-tent definition, just wanted to say the numerical value of "few" seems to vary a lot across the literature

**Audience:**

Yes

**Audience Explanation:**

I've answered Yes above because there does seem to be a sub-population of ML researchers interested in imprinting. There seems to be enough past literature cited here to indicate that.

However, my worry here is that imprinting has a rather narrow use case, when you really need efficiency and you'll tolerate a performance hit to get it. I don't think the present paper does enough to remind readers how this method would compare to *actually adapting parameters to the target task*.

For example, in Fig 6, the ceiling of all imprinting methods for CIFAR-10 with resnet18 is 80% accuracy. Brief scouting online suggests that fine-tuning resnet18 on CIFAR-10 would get closer to 95% accuracy, a notable leap (https://huggingface.co/edadaltocg/resnet18_cifar10). I'd imagine similar gains are possible on other datasets tested here.

I think readers deserve to see explicitly what the accuracy-efficiency tradeoff is for these methods. I'm sure this present method is less expensive, but how much so? Won't most application be tolerant of a few hours of training time?

I'd also appreciate more citations and discussion of real applications the authors have in mind, where the efficiency of imprinting is needed.

**Broader Impact Concerns:**

Nothing specific is concerning here.

**Claims And Evidence:**

No

**Claims Explanation:**

I'll answer "no" about the *present* draft, but I look forward to reading an improved version... I do think a "Yes" might be possible here.

Strengths:

* Framework seems to cover a swath of existing literature
* Coverage of many datasets feels comprehensive
* Coverage of many architectures (2 resnets, 2 transformers) feels comprehensive
* Use of "critical difference" diagrams and attention to significance of experimental results is welcome

Areas to improve:

* within aggregation, the choice of *distance function* used for prediction should be bigger part of framework
* neural collapse ideas seem disconnected
* math details of *variability collapse* are missing


## C1: distance function should be part of AGG framework

To claim that IMPRINT is a general framework, I think you need more attention to aggregation step.

The 'max' aggregation focuses on maximum inner product, which seems equivalent with Euclidean distance *when L2 normalized*. However, in general, it seems like the two key axes in the design space are:

* what distance function is used (Euclidean, Mahalonobis, manhattan, etc.)
* how you aggregate votes from the top m neighbors (uniform, weighted by distance, etc.)

In short, the current AGG step is a bit limited in scope. More comprehensive discussion seems warranted.

## C2: Neural collapse ideas seem disconnected

The abstract claims that the method "determines proxies through clustering motivated by the neural collapse phenomenon". The intro later says there "exists a relationship between a measure of neural collapse and the success of imprinting."

However, I don't see any figure directly connecting the NC numerical measure to classifier success. Fig 11 just shows NC as a function of d, an artificial way of assembling a dataset to get more modes within a class. Fig 12 shows accuracy as a function of k, but doesn't directly connect to NC.

More importantly, it seems a bit tautological to say that if a given class is embedded in more well-separated clumps, then k-means with larger k will be more effective for a nearest-proxy classifier. I don't feel that the "neural collapse" angle adds too much to the discussion here... what am I missing? To ask another way, suppose I measure a specific NC value for a new dataset .... can I avoid a grid search over the k in k-means by directly guessing what k is useful based on the NC value?



## C3: variability collapse metric under-defined

Looking at Eq 1, how do you compute $\Sigma_W$ (within-class covariance) and $\Sigma_B$ (between-class covariance) given a finite dataset of images and class labels?

I know how to compute a within-class covariance matrix for a specific classes (e.g. all dog images) ... but how do I get one $\Sigma_W$ that represents all classes? Do you just average over the each class's matrix? sum?

The between-class covariance matrix calculation should also be clarified, ideally with a concrete formula.

**Requested Changes:**

* 1) better help reader understand the applications you have in mind, where efficiency of imprinting is needed because last-layer or all-layer fine-tuning is not possible, and even storing 'all' training embeddings may be difficult. Surely if you can store all the training images, a little extra space isn't too hard to find? Compared to an image, the embedding might be just 5-10% of the storage, or less!

* 2) help reader understand tradeoffs (acc vs efficiency) of imprinting vs. fine-tuning, at least somewhere in the paper

* 3) Edits to address C1, C2, and C3 issues above


### Presentational requested changes

In Fig 1, I suggest separating the "training time" and "test/prediction time" aspects of the framework into separate diagrams... it is tempting to read the current diagram left to right as what happens at prediction time, which is conceptually inaccurate.

More substantial limitations section in Sec. 6 is needed, addressing differences between gradient-based transfer learning and this method (e.g. accuracy limitations). There are probably several other key limitations that could be listed here.

---

> ### Author Response · Authors · 2025-09-02
> **[1/2]**
>
> We are very grateful for the topics that reviewer 81QZ raised and for acknowledging our insights and contributions. Especially **C2** sparked a lengthy internal discussion that in turn improved the paper substantially by making the NC claims more precise and better connected. Below, we comment on each aspect in detail.
>
> Due to the breadth of the reviewer’s comments, we split our answer into two parts to stay within the 5000-character limit.
>
> **C3: Variability collapse metric under-defined.**
> We thank the reviewer for pointing out the lack of detail in section 3.2 _Quantifying Neural Collapse_. In the revised version, we include the following definitions. Given a finite dataset containing $C$ classes with (for simplicity) $n$ samples per class, we use its $l$-dimensional embeddings $\{h_{k,i}\}_{1\le k \le C,\; 1 \le i \le n}$ produced by a foundation model to define
> - global feature mean $h_G \coloneqq \frac{1}{Cn}\sum_{k=1}^{C}\sum_{i=1}^{n} h_{k,i}$
> - class means $\bar h_k \coloneqq \frac{1}{n}\sum_{i=1}^{n} h_{k,i}$
> - within-class covariance matrix $\Sigma_{W} \coloneqq \frac{1}{Cn}\sum_{k=1}^{C}\sum_{i=1}^{n}\bigl(h_{k,i}-\bar h_{k}\bigr)\bigl(h_{k,i}-\bar h_{k}\bigr)^{\top}$
> - between-class covariance matrix $\Sigma_{B} \coloneqq \frac{1}{C}\sum_{k=1}^{C}\bigl(\bar h_{k}-h_{G}\bigr)\bigl(\bar h_{k}-h_{G}\bigr)^{\top}$
>
> and compute $\mathcal{NC}1 = \tfrac{1}{C}  \mathrm{trace}(\Sigma_{W} {\Sigma^{+}_{B}}),$ where $^{+}$ symbolizes the pseudo-inverse.
>
> **C2: Neural collapse ideas seem disconnected.**
> We highly value the reviewer’s criticism. In fact, we realized that our original phrasing -- "a higher $\mathcal{NC}{1}$ score indicates the benefits of using a higher number of proxies" (Fig. 12) -- was too imprecise or even an overstatement. For example, in the CombiDigits dataset, $\mathcal{NC}{1}$ is very high, yet the effect of using $k>3$ proxies is negligible.
> Our refined observation is that once intra-class variability ($\Sigma_W$) exceeds inter-class variability ($\Sigma_B$), i.e.  becomes larger than $1$, mean imprinting ($k=1$) is no longer sufficient. This is the meaningful “_relationship between a measure of neural collapse and the success of imprinting_” that we can draw. It is clearly visible for CIFAR-10 on the vit_b_16 and swin_b models, both of which yield $\mathcal{NC}_{1} < 1$ and very flat curves over $k$ in Fig. 12. This still supports our abstract’s claim: “_This variant determines proxies through clustering motivated by the neural collapse phenomenon._”
> To highlight this connection more clearly, in the revised version we adapt Figs. 11 and 12 so that they can be shown next to each other and emphasize the $\mathcal{NC}_{1} = 1$ threshold. In other words, given a new dataset and its $\mathcal{NC}_{1}$ value, knowing that it exceeds $1$ already predicts that mean imprinting is far from optimal and can be improved by simple methods such as $k$-means with $k>1$.
> Apart from that, as far as we know, no metric exists that allows direct prediction of the optimal $k$ from a single $\mathcal{NC}_1$ value. The literature emphasizes that this requires evaluating multiple candidate values and applying heuristics, cross-validation, information criteria, or methods like the gap statistic -- each providing relative, not absolute, guidance. Thus the larger the $\mathcal{NC}_1$ -- i.e., the more intra-class variance compared to inter-class variance -- the more critical it becomes to examine intra-class structure via explicit modeling (e.g., varying $k$), rather than expecting a one-to-one inference from $\mathcal{NC}_1$ alone.
>
> **C1: Distance function should be part of AGG framework.**
> We thank the reviewer for this important addition. Defaulting to the `Euclidean` distance is in fact a limitation, which we already removed by adding a new parameter `aggregation_distance_function` to our framework, allowing for `Euclidean`, `Cosine`, `Manhattan`, and `Malahanobis` in the `mnn`aggregation. We are currently running experiments and will provide results shortly. We expect there to be only minor differences and `Euclidean` to remain on top. However, we have already observed that `Mahalanobis` is markedly slower due to covariance estimation/inversion (see **accuracy–efficiency trade-off** below).
> Likewise, we extended the aggregation of votes by adding a new parameter `aggregation_weighting` which can take `uniform` or `distance` (where the latter was the only option before). New experiments are in progress and results will be added. We note that more elaborate voting (e.g., learned top-k attention or entropy-based filtering) is conceivable, but we consider such mechanisms beyond the minimalist imprinting paradigm and thus defer them to future work.

---

> > ### Author Response · Authors · 2025-09-02
> > **[2/2]**
> >
> > **Comparison to methods that adapt parameters.**
> > We thank reviewer 81QZ for emphasizing the importance of comparisons to approaches that explicitly adapt parameters. In response to reviewer n2pB, we have expanded our analysis to include gradient-based methods (see our comment from 20 Aug 2025, 13:29).
> >
> > **More about accuracy-efficiency tradeoff.**
> > We also thank the reviewer for pointing out the need to better quantify the accuracy–efficiency trade-off. To address this, we are rerunning all experiments on a common CPU to obtain consistent run times (in seconds). This will allow us to present clearer results and demonstrate the efficiency gains of our method relative to gradient-based methods.
> >
> > **More citations and discussion of real applications, where the efficiency of imprinting is needed.**
> > We agree with the reviewer that in many practical applications, accuracy is prioritized over computational efficiency, as high-quality predictions or decisions are often more valuable than speed or energy savings -- and thus more motivation is indeed required here.
> > Efficiency becomes a critical requirement in scenarios where computational resources are severely constrained. A prominent example arises in the chemical and polymer processing industries, where edge devices must meet strict safety standards (ATEX, IECEx) and avoid power cables, making battery-powered edge compute essential. Under these conditions, algorithmic efficiency is paramount, as it directly extends device lifetime and ensures reliable operation. Efficient methods are particularly valuable for tasks such as predictive maintenance, image-based quality control, and regression approaches for yield optimization.
> > In the introduction of our work, we mention a “plethora of studies” using imprinting. Among others, we cite Zhu et al. (_Weight Imprinting Classification-Based Force Grasping With a Variable-Stiffness Robotic Gripper_, [IEEE](https://ieeexplore.ieee.org/document/9351747)) and Passalis et al. (_Hypersphere-Based Weight Imprinting for Few-Shot Learning on Embedded Devices_, [PubMed](https://pubmed.ncbi.nlm.nih.gov/32287012/)), two examples of real applications: the former for vision-based robotic force grasping with a variable-stiffness gripper that can safely handle both fragile and heavy objects by rapidly adapting to novel categories without retraining, and the latter tailored for edge-embedded devices lacking training support — cases where (re)training for hours is not an option. The former is one example of the Continual Learning setting, in which in practice it might often be the case that retraining is not feasible. In the revised draft, we elaborate on this in more detail in the introduction. In the revised draft, we elaborate in the introduction about this in more detail.
> > In the related work section, we explicitly mention applications of imprinting to object detection, multi-label classification, semantic segmentation, and more. We acknowledge, however, that the robotic gripper is the only concrete real-world application we originally highlighted, and that we did not sufficiently stress its importance.
> > In the revised version, we extend the discussion of imprinting by focusing more on real-world applications, such as:
> > - Google’s Coral Edge TPU, which supports an _ImprintingEngine_ API to add new classes from only a few examples, while maintaining existing classification without recompiling the model ([Coral AI](https://coral.ai/docs/edgetpu/retrain-classification-ondevice))
> > - The very recent FSID (_Few-Shot Imprinted DINO_), which calculates class prototypes via weight imprinting of spectrogram embeddings from IMU/EMG gait data, enabling classification in low-data regime Human Activity Recognition (HAR) tasks ([Nature Scientific Reports](https://www.nature.com/articles/s41598-025-04323-7))
> >
> > **Presentational requested changes and other minor issues.**
> > We thank the reviewer for these helpful comments.
> > - We have updated Figure 1 to more clearly separate "training" and "test" time.
> > - We have extended the discussion of limitations by explicitly acknowledging the remaining gap between imprinting and gradient-based transfer learning methods, and by highlighting that the choice of k still requires heuristic or empirical selection rather than direct prediction.
> > - We have rephrased the sentence "_Normalization ensures that all class weights contribute equally to the output, making it crucial for a well-performing imprinting method._" to "_Normalization ensures that all class-proxy vectors have the same magnitude, preventing differences in vector norms from disproportionately affecting classifier predictions._"
> > - Regarding the term "few-shot", while for MNIST and FashionMNIST only 50 samples (across all four foundation models) are enough to improve k>1 means imprinting, we agree that our usage was misleading. We will instead adopt the term _“low-data regime”_ (in the contributions, Section 5.2 title, and the conclusion).

---

> > > ### Author Response · Authors · 2025-09-10
> > > **Additional Comment [3/3]: Extended AGG analysis and timing considerations.**
> > >
> > > **Extended AGG analysis and timing considerations.**
> > >
> > > We thank reviewer 81QZ for highlighting the need for a more comprehensive treatment of aggregation (`AGG`). Here are the results with the new `aggregation_distance_function` and `aggregation_weighting` parameters. As expected, these variations have negligible impact on accuracy, with differences well within statistical noise. In the table below, `GEN=all` and `AGG=5-nn`.
> > > | aggregation_distance_function | aggregation_weighting | Avg task acc. % |
> > > | - | - | - |
> > > | cosine | distance | 93.77 |
> > > | manhattan | distance | 93.84 |
> > > | manhattan | uniform | 93.78 |
> > > | euclidean | distance | 93.74 |
> > > | cosine | uniform | 93.67 |
> > > | euclidean | uniform | 93.67 |
> > >
> > > This supports the view that, beyond the choice of $m$, the precise aggregation variant is of secondary importance for imprinting. In the revised draft, we will include these findings for completeness.
> > >
> > > Regarding `Mahalanobis` aggregation, we terminated experiments early once it became clear that runtime was prohibitive: evaluations for `GEN=mean` (i.e., only 1 proxy per class) averaged at ~45s already, compared to ~2s for `Euclidean`.
> > >
> > > Finally, we reran all experiments on the same hardware to obtain consistent timing baselines. We found <10% variation across seeds and report the following numbers (averaged across all tasks):
> > > | GEN | runtime (s) |
> > > | - | - |
> > > | mean | 2.651 |
> > > | 20-means | 10.676 |
> > > | ls | 1.130 |
> > > | 20-ls | 181.785 |
> > >
> > > Interestingly, least-squares (`ls`) turns out faster than ($k$)-means, likely because it processes all data in a single closed-form step instead of sequential clustering iterations. While this undercuts our earlier intuition, we consider it not detrimental to our overall claim: `ls` remains the "optimal" analytic closed-form baseline derived from gradient optimization without iterative training and does not support sequential class handling, which is precisely where imprinting excels (e.g., continual learning or edge-device scenarios).

---

> > > > ### Comment · Reviewer_81QZ · 2025-10-15
> > > > **Reply to Authors on Response 2 of 2**
> > > >
> > > > ## RE request for comparison/discussion of gradient-based methods
> > > >
> > > > > In response to reviewer n2pB, we have expanded our analysis to include gradient-based methods (see our comment from 20 Aug 2025, 13:29).
> > > >
> > > > Unfortunately, I can't see this reply to reviewer n2pB.
> > > > Perhaps it is not globally visible to all reviewers?
> > > >
> > > > I also don't see any revised material in the PDF, the paper seems to be the same from the original submission.
> > > >
> > > > ## RE improved discussion of target applications
> > > >
> > > > Thanks, I appreciate these discussions. I think they will substantially strengthen the paper for its target audience.
> > > >
> > > > ## RE presentational changes
> > > >
> > > > I'm glad to hear of the revisions. I do wish I could have seen them in a revised PDF (in other TMLR review processes, there's been a revised PDF submitted by the authors during response period).
> > > >
> > > > Especially the revision to Fig 1 (and the changes to Fig 11 and 12 mentioned elsewhere) would have been good to have reviewers check.

---

> ### Comment · Reviewer_81QZ · 2025-10-15
> **Thanks for your response [1 of 2]**
>
> I appreciate the authors' careful point-by-point response here. A few quick thoughts on the response marked "1/2" (1 of 2).
>
> ## C3: Variability collapse metric under-defined
>
> I agree if the provided equations in the response above are added to the revised paper, I'll consider the major issue behind C3 resolved. It is now possible to -- in a reproducible way -- compute this metric.
>
> That said, I would encourage authors in future to submit a revised PDF version of the manuscript with such changes during the revision/rebuttal phase. Ideally, I'd like to see the final text of the paper before fully signing off on the resolution.
>
> ## C2: Neural collapse ideas
>
> I appreciate that the authors have
>
> > once intra-class variability ($\Sigma_W$) exceeds inter-class variability ($\Sigma_B$), i.e. becomes larger than $1$, mean imprinting ($k=1$) is no longer sufficient
>
> My own summary of this statement is:
>
> * when NC <=1, mean imprinting with k=1 is "sufficient"
> * when NC > 1, this is when k > 1 is needed
>
> I'm concerned, though, that this claim doesn't seem supported by my reading of Fig 12. Consider the 10-in-1 lines (teal color) in Fig 12: it does not seem "flat" incases where NC < 1. For vit, NC is 0.6 by Fig 11, below the 1.0 threshold yet in Fig 12, the teal line goes from 87% to above 90% as k increases. For swin (where NC is maybe 0.45 by Fig 11, again below the threshold of 1.0), the teal line goes from 91% to 94% as k increases.
>
> So I think even in Fig 12, we have some contradictory evidence that k=1 is "enough" for NC far below 1.0.  I do agree that some other lines do appear "flatter" in Fig 12 for the vit or swin models, but I worry this is because these models are better at k=1 than resnets, so there is just less room to grow.
>
> I'm thus rating my concern C2 as still unresolved. I think the NC metric remains potentially interesting, but I'm not sure how it helps practitioners apply this framework... I think we'd still recommend trying various k values if you care about improving your overall accuracy.
>
> I appreciate the authors' idea to put Fig 11 and Fig 12 side-by-side... I'd suggest even more drastic changes, to perhaps directly show something like "gain in accuracy for k = 5 over k =1" versus NC, to see whether some kind of trend makes sense. Currently, these figures are neat but require the reader to do a lot of work to map between an NC value and an accuracy.
>
> ## C1: Distance functions
>
> Thanks for adding experiments, I look forward to seeing results.

---

> ### Author Response · Authors · 2025-10-17
> **Reply to Reviewer's Responses 1 and 2**
>
> We thank reviewer 81QZ for their additional comments. Below, we address each remaining point.
>
> **Revised version**
>
> First of all, we can totally understand the reviewer's desire to see the improvements incorporated into a new PDF and we are sorry about the confusion that this revised version is still missing. It was unclear to us, that we were expected (or even allowed) to upload a revised version yet, before we hear from the Action Editor (AE). In the meantime, we have asked the AE whether we can/should upload a revised version, and once confirmed, will do so as soon as possible, including all the improvements we have received from the reviewer(s). We hope that the reviewers will then still be able to check if they are satisfied with how their improvements were incorporated into the paper.
>
> **C2: Neural collapse ideas**
>
> We agree with the reviewer who rightfully points out that for the tested ViTs, NC on our relabelled ImageNet variants is less than 1, and yet increasing k slightly increases the imprinting accuracy.
> Together with the reviewer's idea of an additional plot that more clearly shows the gain in accuracy by using k>1 over NC, however, we find that (see the anonymously uploaded plot here: [https://upload.imgshare.cc/images/3vb343x8.webp](https://upload.imgshare.cc/images/3vb343x8.webp))
> indeed there is a clear change in behavior around NC=1 such that it seems valid to us to state that
>
> > mean imprinting ($k=1$) is no longer sufficient, once intra-class variability ($\Sigma_W$) exceeds inter-class variability ($\Sigma_B$), i.e. once NC becomes larger than $1$
>
> To be more precise, **our experiments indicate that once NC exceeds 1, i.e., when intra-class variability ($\Sigma_W$) surpasses inter-class variability ($\Sigma_B$), using more than a single proxy per class ($k>1$) yields a clear performance gain over mean imprinting ($k=1$)**.
>
> Also, we agree that from a practical perspective choosing k based on pure greedy search with validation data (or as part of any AutoML pipeline) is still a valid option. However, our analysis provides insights into the underlying mechanism and could guide the selection process for k.
>
> In a final version, we will add this plot in a new figure, but also keep the other existing two, since they convey other messages (general NC behavior, "tipping point" for k=d, etc.) as well.
>
> Altogether, we are now even more convinced that NC offers valuable insights into the underlying mechanisms and performance of different imprinting strategies.
>
>
> **C1: Distance functions**
>
> The desired results can be found above in our reply titled "Additional Comment [3/3]: Extended AGG analysis and timing considerations".
>
> **RE request for comparison/discussion of gradient-based methods**
>
> As our answers to other reviewers seem be invisible to reviewer 81QZ, we copy over the relevant insights from our reply to reviewer n2pB:
>
>
> > **Gradient descent comparison.**
> We think this is an important suggestion by the reviewer. We provide such results in appendix A.5 focusing on the multi-modal ImageNet scenario, showing that `k-means` is competitive with analytical least-squares initialization (which corresponds to the exact optimum that gradient descent would converge to under squared error,) with access to all classes at once (hence using cross-class statistics). More precisely, we find that "with increasing multi-modality d, multi-proxy methods (`k-means`, `k-least-squares`) generally outperform single-proxy methods (`mean`, `least-squares`), and that the `k-means` imprinting scheme is competitive with least-squares approaches" (Figure A.8).
> Furthermore, we ran new experiments on all the tasks described and properly evaluated in/for section 5.1 to be able to provide an updated Table 1, now also including a gradient-based method (optimal least squares (LS) weights):
>
> | Work | NORMpre | GEN | NORMpost | NORMinf | AGG | Avg. acc. % |
> | - | - | - | - | - | - | - |
> | Qi et al. (2018) | L2 | mean | L2 | L2 | max | 86.79 |
> | Hosoda et al. (2024) | none | mean | quantile | none | max | 82.90 |
> | Janson et al. (2022) | none | mean | none | none | 1-nn | 86.64 |
> | Ours | L2 | 20-means | L2 | L2 | max | 91.06 |
> | LS weights | none | ls | none | none | max | 94.54 |
>
> > (Note that the other numbers also slightly changed, as we realized that we missed the CIFAR-10 accuracies and rankings in accumulation for that (and the other) tables!)
>
> > However, it should be kept in mind here that this is not an apples-to-apples comparison, as calculating LS weights uses cross-class statistics and generally requires more compute.
> Still, this suggestions helps us make the scope clearer. We favor showing these results in section A.5, as they otherwise deviate from our main focus on imprinting, but we are open to discuss this manner further.

---

> > ### Comment · Action_Editor_SFfv · 2025-10-17
> >
> > Dear authors,
> >
> > regarding your question:
> >
> > > First of all, we can totally understand the reviewer's desire to see the improvements incorporated into a new PDF and we are sorry about the confusion that this revised version is still missing. It was unclear to us, that we were expected (or even allowed) to upload a revised version yet, before we hear from the Action Editor (AE).
> >
> > please refer to TMLR's author guide: ("Then reviewers will be assigned to read and review the manuscript. An open-ended rebuttal, discussion, and revision phase will allow authors to interact with reviewers and update their paper.", https://jmlr.org/tmlr/author-guide.html).
> >
> > So please feel free to still update your submission now.
> >
> > -AE

---

### Review · Reviewer_cfTm · 2025-08-28

**Summary Of Contributions:**

The work introduces a framework to analyze weight imprinting in transfer learning, splitting it into generation, normalization, and aggregation steps. The proposed framework is conceptually clear and provides a common view for previous methods. Experimentation is comprehensive: different architectures, datasets and scenarios. They also link the success of imprinting with the neural collapse phenomenon, which provides a theoretical angle.

**Audience:**

Yes

**Audience Explanation:**

The proposed framework unifies weight imprinting methods and connects them to neural collapse, which are relevant for researchers in transfer learning and related areas.

**Claims And Evidence:**

Yes

**Claims Explanation:**

The claims are well supported by extensive experiments and improvements over prior methods, although broader baselines and scalability analysis would strengthen the evidence.

**Requested Changes:**

- Try or discuss the computational cost and scalability to larger and more challenging datasets, including potential bottlenecks in the framework steps.
- Consider at least a discussion of extensions to other modalities and tasks.

---

> ### Author Response · Authors · 2025-09-02
>
> We sincerely thank reviewer cfTm for bringing up these points and for recognizing our insights and contributions. Below we respond to the requested clarifications.
>
> **Computational cost and scalability.**
> We thank the reviewer for raising the issue of computational cost and scalability. Our framework inherits the efficiency of imprinting, and the main additional step we introduce in the best-performing method that we find -- k-means clustering --, as well as the other steps, scale well:
> - **GEN (k-means clustering):** In our implementation (sklearn.cluster.KMeans with algorithm="lloyd"), the complexity is $O(nklt)$ with $n$ samples of dimension $l$, $k$ clusters, and $t$ iterations, i.e. linear in $n$ for fixed $l$, $k$ and $t$. While k-means is not "free", it has a modest computational footprint, both assignment and update steps parallelize naturally, and convergence typically requires only a few iterations.
> - **NORM (L2 normalization):** Both pre- and post-normalization are simple vector operations with cost $O(nl)$, negligible compared to GEN or AGG.
> - **AGG (m-nearest neighbor aggregation):** At inference time, each test embedding is compared against all $Ck$ proxies. With $C$ classes and $k$ proxies per class, the cost is $O(Ckl)$ per sample. This scales linearly in the number of proxies and remains small compared to a forward pass through the backbone.
>
> Because all steps scale linearly with dataset size or proxy count and parallelize easily, we do not see scalability bottlenecks in practice. This stands in clear contrast to gradient-based transfer, which requires repeated forward passes and even closed-form least-squares weight computation involves $O(nl^2 + l^3)$ operations for covariance estimation and matrix inversion. We will add this discussion in an efficiency section where we also add results from efficiency-accuracy tradeoffs (see other replies).
>
> **Extensions to other modalities and tasks.**
> As noted in our response to reviewer 81QZ, imprinting is particularly relevant in settings where retraining is infeasible (e.g., robotics, embedded devices, or edge computing under strict safety constraints). Beyond vision classification, imprinting has been applied to object detection, multi-label classification, and segmentation. More broadly, the same GEN–NORM–AGG framework can extend to non-visual modalities such as audio or sensor data, where embedding-based classification with prototypes is already common. Very recently, [FSID (_Few-Shot Imprinted DINO_)](https://www.nature.com/articles/s41598-025-04323-7) applied weight imprinting to IMU/EMG spectrogram embeddings for human activity recognition, further demonstrating its suitability in low-data regimes.
> Finally, continual learning is a natural extension. Imprinting supports incremental addition of new classes without retraining, and prior work already demonstrates its use in continual settings (e.g., [Janson et al., 2022](https://arxiv.org/abs/2210.04428v2)). In practice, this is one of the most promising directions: in robotics or industrial monitoring, new categories or failure modes often arise dynamically, and efficient, non-destructive integration of such classes is essential. We will emphasize this more explicitly in the revised draft.

---

### Decision · Action_Editor_SFfv · 2025-11-20

**Recommendation:** Accept with minor revision

**Additional Comments:**

I would like to ask the authors to address some remaining reviewer concerns. I believe that these could be easily addressed during a minor revision of the paper:

- First, the authors fail to remind the reader of the tradeoffs of their imprinting strategy compared to the “exact” solution of adapting the last or internal layers using gradient descent. Reviewers n2pB and 81QZ discuss this issue with the comparison to gradient based methods, which the authors replied to (https://openreview.net/forum?id=duU11BnQ3Y&noteId=UUHFiB32XC); cfTM also notes that better baselines comparisons are required. The comparison outlined in the discussion phase should be added to the *main* paper, in Table 1 (94.54%) and contextualized. I agree with reviewers that it is important to position the method with respect to this “exact” solution; it is clear that there is a tradeoff due to imprinting. I would recommend adding this “exact solution” to other tables as well to contextualize how difficult the respective tasks are (Tables 2-6); it could be visually highlighted, e.g. via gray text, that this is not an apples-to-apples comparison.

- Second, reviewer 81QZ rightfully pointed out the missing empirical connection between the proposed NC metric and the claim that higher NC values necessitate using k>1. The authors revised Figures 11 and 12 to clarify this relationship. While the reviewer did not evaluate the updated figures after submitting their final recommendation, I re-examined the revised section. What remains, is the reviewer’s explicit request to quantify the relationship rather than rely solely on visual inspection. Please add an analysis reporting the correlation between NC and downstream performance gains, together with an appropriate statistical test, to back the claim of a “clear linear relationship.”

Some additional minor issues I noticed during re-reading the paper:

- The CD diagrams are great for significance testing, but not described in the methods.
- The abstract currently reads “It has been reinvented several times, but not systematically studied”. I would challenge this exact claim and recommend to tune it down, even if this was not remarked during review; it seems that there were still conceptual differences between these studies which then formed the basis for the systematic investigation in the paper.
- Type in Figure 3, o1 appears twice
- A %-sign is missing in Table 4
- Given the amount of experiments conducted, it would be a valuable addition to report confidence intervals/standard deviations also in the tables.

**Audience:**

Yes

**Audience Explanation:**

The paper fits into the scope of TMLR. I agree with the reviewers that especially the conceptualization of common imprinting frameworks and their systematic study falls very well into the scope of the journal.

**Claims And Evidence:**

Yes

**Claims Explanation:**

The authors study “imprinting”, a method for fast adaptation of neural networks for classification. For imprinting, embeddings of a foundation model are aggregated into one or more representative prototypes for each class. At inference, embeddings are normalized, compared to these prototypes, and an aggregation step selects the predicted class.

After breaking the methodology into the aforementioned computational steps, the authors organize existing literature on imprinting in terms of different parameterizations of these steps, and then systematically evaluate. As 81QZ and cfTm note, the authors do a good job at contextualizing their work in the literature in this way. I also agree with the reviewer that this review and categorization of the literature indicates that “there does seem to be a sub-population of ML researchers interested in imprinting”. The experimental methodology and statistical procedures are also positively received by reviewers, though cfTm notes that the set of baselines could be broadened further.

In the proposed imprinting framework, the number of prototypes to store for each class can be selected as k=1 or k>1. The authors revisit the concept of neural collapse as described by Papyan et al. (2020) (“convergence of the last-layer weight vectors to class means”) and (plausibly) hypothesize that models exhibiting stronger neural collapse should require only a single prototype per class (k=1; mean imprinting), whereas models that do not show neural collapse would benefit from aggregating a larger number of prototypes.

The authors introduce a metric for quantifying neural collapse, and claim that this metric can serve as a proxy for the required number of samples for imprinting. This connection is investigated later on in the paper, now in particular in Figures 11 and 12. Reviewer 81QZ had remaining concerns about how this connection was presented and discussed during the rebuttal phase, particularly regarding the rigour of the analysis.

Following the discussion phase, two reviewers leaned toward acceptance, and the third indicated that the paper could become convincing with further revisions. Because the authors, due to a misunderstanding of the process, only uploaded a revised manuscript after decisions were posted, I re-read the updated version to verify whether the remaining reviewer concerns had been addressed.
While the most pressing concerns were addressed, some additional minor revisions are required. In summary, I can recommend the paper for acceptance if these are addressed.

---

> ### Author Response · Authors · 2025-12-03
>
> We genuinely thank the AE for their decision, comments and further feedback. Especially, we are very thankful for them rereading the updated version again in detail to verify if remaining concerns of the reviewers were proprely addressed by us.
>
> We have finished the camera-ready version, though have not uploaded it yet because we are in still in the process of creating a video presentation (which we have to link when handing in the camera ready version). Our plan is to have this ready by next week.
>
> However, we have just uploaded our final version as a revised version here to be able to make sense of the below comments and to receive some final feedback about the minor revisions we have performed.
>
> **Comparison to "exact" solution**
>
> We very much agree with the AE and added an "Oracle" row in our main Table 1 and explain the tradeoff in its caption. However, we have decided against adding this same row and number into all the other tables as we found that to be redundant and, more importantly, misleading. Tables >1 all examine specific parts of the IMPRINT framework (e.g., vary only GEN, only NORM, only AGG, ...), so a constant comparison to the optimal oracle (which does very different things) did not seem meaningful to us.
>
>
> **Quantified neural collapse relationship**
>
> We thank the AE for bringing this up, again. We agree that it still was not clear/quantitive enough. To solve this, we have added a regression line with $R^2=0.36$ in Figure 12 and report the Spearman rank correlation $\rho = 0.82$ ($p<0.0001$) in the caption. With this, we have a quantitive indication of a statistically significant positive association.
>
>
> **About the other minor issues**
>
> - added section 3.3 "Significance Testing with Critical Difference Diagrams" explaining CD diagrams
> - replaced “It has been reinvented several times, but not systematically studied” by "The conceptual differences between studies on imprinting form the basis for our systematic investigation." in the abstract
> - missing %-sign in table 4 and typo in figure 3 fixed, great catches!
> - added stds in all tables (and properly explained in the methods section and the caption of Table 1 that the stds are expectedly very high as the groups are very heterogenous (different datasets and models), which in turn justifies our ranked based aggregation and analysis using CD diagrams)
>
>
> Once again, we would like to sincerely thank the AE and reviewers for making all these improvements to our paper possible through all their detailed and helpful feedback. We are very glad about how the process went!

---

> > ### Comment · Reviewer_81QZ · 2025-12-03
> > **Thanks to the authors and AC for an improved paper**
> >
> > Just wanted to comment here to say I've been able to skim the recent revision from early December, and I think this is a much better paper.
> >
> > **RE new oracle results**: Thanks for adding this! The gray coloring helps the reader know this comparison isn't really "apples to apples" but that nevertheless for increased time and resources better models beyond imprinting are possible. I do appreciate that this comparison is best done in Table 1 to give overall context, and the within IMPRINT detailed comparisons in later tables/figures can be left alone.
> >
> > **RE new Fig 12**: I think this figure is helpful, and I appreciate the recent assessment of the correlation with statistical tests.
> >
> > I am sorry for the lack of replies throughout November. I appreciate the AC's diligence in steering this to a nice resolution for all parties.

---

> > > ### Author Response · Authors · 2025-12-04
> > >
> > > We appreciate that Reviewer 81QZ checked in again and provided feedback to the final revisions. Generally, we are very thankful for all their feedback that significantly improved our work.

---

> > ### Author Response · Authors · 2025-12-05
> >
> > We have just uploaded the camera-ready version, together with links to a video presentation and code to reproduce all our results. If anything remains unclear or further questions arise, we are happy to clarify it.